**Assessment framework to predict sensitivity of marine calcifiers to ocean alkalinity enhancement - identification of biological thresholds and importance of precautionary principle**
Nina Bednaršek[1]*, Hanna van de Mortel[2], Greg Pelletier[3], Marisol García-Reyes[4], Richard A. Feely[5], Andrew G. Dickson[6]
[1]*Cooperative Institute for Marine Ecosystem and Resources Studies, Hatfield Marine Science Center, Oregon State University, 2030 SE Marine Science Drive Newport, OR 97365, USA
[2]HvdMortel Consulting, Utrecht, NL
[3]Washington Department of Ecology, Olympia, 300 Desmond Dr SE, WA 98503, USA (Emeritus)
[4]Farallon Institute, 101 St. Suite Q, Petaluma, CA 94952, United States
[5]NOAA Pacific Marine Environmental Laboratory, Seattle, WA, 98115 USA
[6]University of California at San Diego, Scripps Institution of Oceanography, 9500 Gilman Drive, La Jolla, CA 92093, USA (Emeritus)
*Correspondence to*: nina.bednarsek@oregonstate.edu

**Abstract**

Ocean alkalinity enhancement (OAE), one of the marine carbon dioxide removal strategies, is gaining recognition in its ability to mitigate climate change and ocean acidification (OA). OAE is based on adding alkalinity to open-ocean and coastal marine systems through a variety of different approaches, which raises carbonate chemistry parameters (such as pH, total alkalinity, aragonite saturation state), and enhances the uptake of carbon dioxide ($CO_2$) from the atmosphere. There are large uncertainties in both short- and long-term outcomes related to potential environmental impacts, which would ultimately have an influence on the social license and success of OAE as a climate strategy. This paper represents a synthesis effort, leveraging on the OA studies and published data, observed patterns and generalizable responses. Our assessment framework was developed to predict the sensitivity of marine calcifiers to OAE by using data originating from OA studies. The synthesis was done using raw experimental OA data based on 68 collected studies, covering 84 unique species and capturing the responses of eleven biological groups (calcifying algae, corals, dinoflagellates, mollusks, gastropods, pteropods, coccolithophores, annelids, crustacean, echinoderms, and foraminifera), using regression analyses to predict biological responses to NaOH or $Na_2CO_3$ addition and their respective thresholds. Predicted responses were categorized into six different categories (linear positive and negative, threshold positive and negative, parabolic and neutral) to delineate responses per species. The results show that 34.4% of responses are predicted to be positive (N=33), 26.0% negative (N=25), and 39.2% (N=38) neutral upon alkalinity addition. For the negatively impacted species, biological thresholds, which were based on 50% reduction of calcification rate, were in the range of 50 to 500 µmol/kg NaOH addition. Thus, we emphasize the importance of including much lower additions of alkalinity in experimental trials to realistically evaluate *in situ* biological responses. The primary goal of the research was to provide an assessment of biological rates and thresholds predicted under NaOH/$Na_2CO_3$ addition that can serve as a tool for delineating OAE risks, guiding and prioritizing future OAE biological research and regional OAE monitoring efforts and communicate the risks with stakeholders. This is important given the fact that at least some of the current OAE approaches do not always assure safe biological space. With 60% of responses being non-neutral, a precautionary approach for OAE implementation is warranted, identifying the conditions where potential negative ecological outcomes could happen, which is key for scaling up and avoiding ecological risks.

## 1. Introduction

Anthropogenic carbon dioxide ($CO_2$) emissions have increased at an unprecedented rate and have contributed to global climate change and negative ecological and biogeochemical impacts in the oceans (Feely et al., 2004; Gattuso et al., 2018), to the extent of crossing six different planetary boundaries (Richardson et al., 2023). Oceans play a crucial role in attenuating the increase in atmospheric $CO_2$ through the absorption of the excess atmospheric $CO_2$ of roughly a quarter of anthropogenic carbon dioxide ($CO_2$) emissions, drawing down around 2–3 Pg C $yr^{-1}$ in recent decades (Friedlingstein et al., 2022). However, without substantial $CO_2$ emissions abatement and $CO_2$ removal strategies, profound repercussions on climate, extreme weather events, and socioeconomic implications will follow. Ocean-based $CO_2$ removal and sequestration strategies (broadly referred to as marine CDR) are among the proposed CDR approaches that remove $CO_2$ and store it for geologically relevant times (National Academies of Sciences, Engineering, and Medicine, 2021). These mCDR approaches only complement $CO_2$ emission reductions and contribute to the portfolio of climate response strategies needed to meet the global goal of limiting warming to well below 2°C as established by the Paris Agreement. Various mCDR approaches have unique benefits and costs but differ in their value depending on their state of implementation, and whether they act globally and/or locally (Oschlies et al., 2023).

Ocean alkalinity enhancement (OAE) has the potential to mitigate climate change through increasing ocean uptake of $CO_2$, while simultaneously reversing ocean acidification (OA) and improving marine habitats. Despite mostly being in the concept stage, OAE is viewed with a high level of confidence as to its effectiveness: medium on environmental risk, but low on the underlying knowledge base (Eisaman et al., 2023; Gattuso et al., 2021; National Academies of Sciences, Engineering, and Medicine, 2021). One of the major concerns about OAE is large uncertainties in both short- and long-term OAE outcomes related to potential environmental impacts of OAE (Kheshgi, 1995; Bach et al., 2019), especially if OAE were to induce novel conditions in the marine systems that are outside the range of the natural variability, exposing organisms to conditions not experienced in their evolutionary history. The outcome of OAE as a successful climate strategy depends on a thorough and advanced understanding of the impacts of OAE implementation while avoiding or minimizing negative biological effects.

## 1.1 Leveraging ocean acidification research on marine calcifiers


Increased $CO_2$ uptake, which initially is absorbed by the ocean as dissolved $CO_2$, causes a decline
in pH, shoaling of the saturation state horizon ($\Omega_{ar}$) and reduced carbonate ion amount content in
a process termed ocean acidification (Feely et al., 2004), causing negative consequences to marine
biota, especially marine calcifiers, the structure and function of the vulnerable marine ecosystem,
and alteration of the carbon cycle. On the other hand, chemical changes induced by OAE are
inherently linked to reversing the OA process: increasing pH, shifting carbonate chemistry
speciation towards lower aqueous carbon dioxide ($pCO_2$) and higher carbonate ion ($CO_3^{2-}$) content,
as well as higher aragonite saturation state ($\Omega_{ar}$). Such changes could either be within the ranges
of the variability of the natural systems to which species are acclimatized, or outside them, creating
novel conditions for which species might not have developed suitable acclimation strategies. As
such, the biological outcomes are, due to their complexity, highly unpredictable.
Scientific progress over the past 30+ years of OA research has brought substantial insights into the
biological effects, with the most fundamental outcome being that calcifying organisms would be
primarily affected (Riebesell and Gattuso, 2015), with the calcification process being one of the
most susceptible pathways, underpinned by species differences in calcification mechanisms (Ries
et al., 2009; 2011; Bach et al., 2013; 2015; Leung et al., 2022). However, OA focused heavily on
investigating biological effects on the higher acidity range of the carbonate chemistry conditions
predicted under future scenarios and most of the studies focused on manipulating the level of $pCO_2$
rather than alkalinity. This resulted in poor understanding of the biological effects at the higher pH
end of the carbon chemistry range (Renforth and Henderson, 2017). Some biological inferences
can be made based on the understanding of the physiological mechanisms underlying the
calcification mechanisms (Bach et al., 2019), but such insights are not adequate to provide
sufficient understanding. Despite the lack of biological data at the upper ranges of pH and $\Omega_{ar}$, this
study builds on the premise that previous OA studies could be leveraged for assessment of
biological responses under OAE. Comparative experimental work, meta-analyses, and the
threshold work (Kroeker et al., 2013; Leung et al., 2022; Bednaršek et al., 2019; 2021b,c) have
indicated that even very diverse responses can be grouped into categorical responses.
Calcification is a primary pathway through which organismal sensitivity to OA is expressed. It is
directly involved in growth and (abnormal) development across most marine calcifiers, and it
indirectly influences susceptibility to predation. As such, calcification can serve as an early
warning indicator of stress, while also playing a crucial role in the ecological success of numerous
marine calcifiers. Studies have shown that the thresholds for calcification occur at similar pH and
saturation state ($\Omega$) values as those affecting energy metabolism processes (Lutier et al., 2022;
Bednaršek et al., 2019; 2021b,c). Furthermore, calcification is directly linked to carbon export,
which has significant biogeochemical implications that may influence the efficiency of OAE. This
study aims to systematically assess the calcification responses of various species under predicted
conditions following carbonate-based OAE compound addition.
**1.2. Complex carbonate chemistry changes induced by various OAE compounds**
Various OAE compounds added to the water change carbonate chemistry in a multifaceted way
and require complex calculations of a multi-parameter problem. As the values of TA and DIC
change, a variety of other parameters, such as pH, $CO_3^{2-}$, $\Omega_{ar}$, and $pCO_2$ exhibit approximately
linear relationships, with slopes that vary along these lines (see Fig. 1). This means that if TA and
DIC vary in proportion to one another, then the values of these displayed parameters hardly change
at a particular salinity, temperature, and pressure. With TA, DIC and the hydrographic conditions
(salinity, temperature and pressure), one can constrain the carbonate system. Our method requires
us to have *one* variable constraining the entire carbonate system. TA and DIC have the benefit that
they can both be directly measured with high precision and accuracy or calculated from other
carbonate parameters. They are also both directly linked to OAE, as we are enhancing the TA
which then allows DIC to increase over time due to the gradual uptake of atmospheric $CO_2$.
To demonstrate the changes of the carbonate system in the experimental system, Figure 1 shows
the changes in carbonate parameters with the addition of two OAE compounds, i.e. NaOH (solid
line) and $Na_2CO_3$ (dashed line) to seawater. When NaOH is added, only TA increases and when
$Na_2CO_3$ is added, TA and DIC increase at a 2:1 ratio. This results in corresponding changes in pH
(Fig. 1a), $\Omega_{ar}$ (Fig. 1b) and $pCO_2$ (Fig. 1c) and shows how much of a change is required to bring
the system back to equilibrium with respect to the atmosphere.

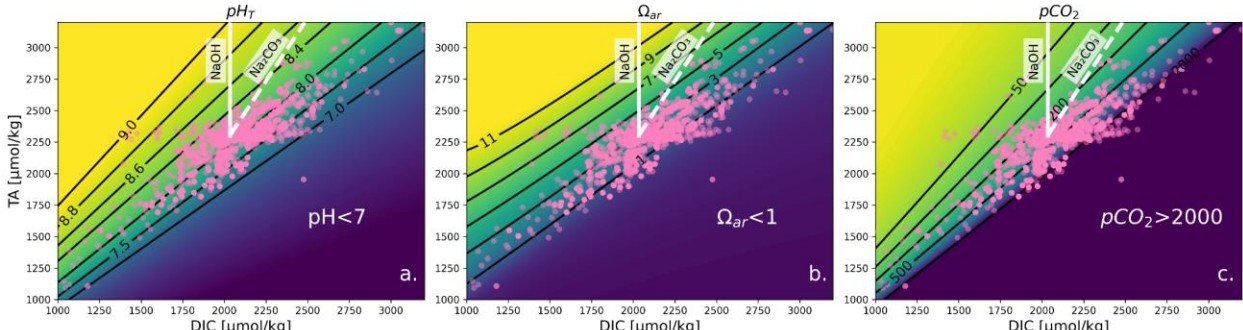


**Figure 1:** *The effect of changes in TA and DIC on the properties of seawater (S= 34.68, T=16°C,*
*[SiO_2] = 50 µmol/kg, [PO_4^{3-}] = 0.5 µmol/kg, TA = 2303 µmol/kg, DIC = 2034 µmol/kg), adapted*
*from Schulz et al. (2023). Pink dots represent experimental TA and DIC data used in our synthesis.*
*Subfigures show pH_T, Ω_{ar} and pCO_2 (in µatm). Calculations were carried out with the Python*
*version of CO2SYS (Humphreys et al., 2022) using the stoichiometric dissociation constants for*
*carbonic acid from Sulpis et al. (2020), for sulfuric acid by Dickson et al. (1990) and for total*
*boron from Uppström (1974). The solid white line indicates the effect of adding NaOH and the*
*dashed white line indicates the effect of adding Na_2CO_3. This grouping of lines can be translated*
*so that its initial position moves elsewhere to visualize different initial conditions. Note that at TA*
*< 1000 µmol/kg and DIC < 500 µmol/kg the isolines are no longer straight when considering Ω_{ar},*
*however, such conditions are rare in the ocean and not widely applicable. The same contour plot*
*utilizing GLODAP data plotted instead of experimental data is shown in Supplemental Figure 1.*

**1.3 Testable conceptual framework based on the existing OA studies**

Based on Ries et al. (2009), calcification responses can be categorized into six categories (Fig. 2):
linear positive or negative response; threshold positive or negative response (exponential fit);
parabolic response; and neutral (no significant) response. We hypothesize that these categories of
responses based on ocean acidification data and delineated by Ries et al. (2009, 2011), could also
be applicable to OAE dosing. For this meta-analysis, we have undertaken three steps: first,
synthesize carbonate chemistry data at regional and global scales to obtain TA, DIC and Ω_{ar}
correlations; second, conduct a literature review and collect available data from OA literature
related to the calcification rate responses across the species of eleven groups of marine calcifiers;
and third, run regression analyses and determine the category of calcification rate response to
TA:DIC, further extending it with addition of NaOH and Na_2CO_3.

The most accurate way of predicting the responses to OAE addition is done based on the
mechanistic understanding of calcification response to specific carbonate chemistry parameter(s).
The hypothesis was that if mechanistic relationships with identified carbonate chemistry driver(s)
are available for species, calcification rate under various feasible OAE scenarios can be predicted
with greater accuracy and lower uncertainty. We further focused on investigating if the empirical
results were consistent with mechanistic calcification predictions for a few selected species for
which the mechanisms were known.

Here, we demonstrate the TA:DIC relationship with calcification rates and show the application
for the TA:DIC thresholds beyond which the responses become negative. Ultimately, we
synthesize which calcifying species or groups are predicted to benefit or lose due to OAE, what
constitutes a species-specific safe operating space related to OAE, and we delineate what
experiments are most urgently needed to fill in critical knowledge gaps before massive OAE field
implementation can be considered.
**2. Methodology**
**2.1 Literature review of data on marine calcification impact by OA**
To assess the impact of OAE on a range of marine calcifiers, we used existing studies on marine
species calcification response to OA that had aligned raw biological (calcification rate) data along
with corresponding carbonate chemistry. We searched within Scopus, Web of Science, and
PubMed and used datasets that were archived in NCEI, OA-ICC and Pangaea. Through personal
correspondence, we have additionally contacted lead authors of the studies whose data are not or
are insufficiently archived. Searches for biological datasets relating to calcification rate and
corresponding carbonate chemistry were conducted through November 2023, encompassing 68
existing studies. The aim was to cover a wide range of calcifying organisms across various
functional groups and 84 species. For several functional groups data was easy to find (algae,
coccolithophores, corals, foraminifera, mollusks and dinoflagellates), so no new studies were
added after 10 to 15 studies were found. Seven studies were found for pteropods, five for
gastropods, four for echinoderms, three for crustaceans and one for annelids. When reviewing the
literature, we included data from the OA experimental studies related to the physical-chemical
parameters (temperature, salinity, TA, DIC) and biological data related to calcification rate.

## 2.2 Use of TA:DIC instead of $\Omega_{ar}$ or pH

Understanding the change in carbonate chemistry upon alkalinity addition is essential for the
biological experimentalists who are conducting biological assessments to report on the effects of
OAE. However, complex changes in the carbonate chemistry induced by alkalinity addition are
not intuitive or straightforward; in fact, they are multi-parameter problems that require complex
carbonate chemistry calculations. Using the TA:DIC ratio is a more practical way of looking at the
impacts of the OAE treatment instead of using a single carbonate parameter because of the high
degree of correlation between TA:DIC and other carbonate system parameters (see Fig. 1).
With TA, DIC and the hydrographic conditions (salinity, temperature and pressure), one can fully
constrain the carbonate system. Our method allows *one* variable constraining the entire carbonate
system. TA and DIC have the benefit that they can both be directly measured or calculated from
other carbonate and physical parameters. They are also both directly linked to OAE, as we are
enhancing the TA which then allows DIC to increase over time due to the gradual uptake of
atmospheric $CO_2$ (Fig. 1 shows the changes in the carbonate chemistry system upon NaOH and
$Na_2CO_3$ addition).
Our focus was on streamlining the process of expressing experimental results and subsequently
reporting responses, with the goal of reducing the multi-parameter complexity into a single-
parameter simplification. This step reduces multiple degrees of freedom into just two, i.e. TA and
DIC, with the ratio allowing us to consider this as a 1-parameter problem. As such, TA:DIC is a
simplistic and convenient way of describing the system, where we only need to understand the
change in TA and DIC ratio, which is feasible for every OAE compound added to the experimental
system. In addition, TA:DIC is also the best approximation for the $CO_3^{2-}$ concentration. The
insights from multiple biological experimental studies show that the $CO_3^{2-}$ concentration is the
representative driver of the calcification process for multiple calcifying groups, although not all,
compared to $\Omega_{ar}$, which represents an empirical approximation based on a number of physical and
chemical parameters. Furthermore, by using TA:DIC we do not have to choose a particular
parameter to describe the changes in calcification. It could also work for the species in which other
parameters drive the calcification, e.g. bicarbonate in autotrophic species, $\Omega_{ar}$ in bivalves and $H^+$
flux in foraminifera. In that way, we standardize all the parameters that would otherwise influence
the carbonate system and come up with a more uniform way to express the experimental
conditions, which would then be useful for easier comparisons among the conducted experiments.
For the ease of comparing TA:DIC with pH and $\Omega_{ar}$, we refer the reader to Supplemental Table 1
and Supplemental Fig. 2.

## 2.3 Experimental biological and biogeochemical data

Based on the collected data, the range of pH and $\Omega_{ar}$, experimental conditions used and their
TA:DIC relationship was determined (Supplemental Fig. 2 and Supplemental Table 1). Most
studies covered pH conditions from 7.5 to 8.5 and $\Omega_{ar}$ from <1.0 to values up to 5.0, with a few
studies increasing pH up to 9 and exceeding $\Omega_{ar}$ of 10. This indicates the potential of leveraging
such experimental studies as a baseline for predictive regression models of biological responses to
a range of $\Omega_{ar}$ conditions, as expected under OAE studies.
Once the biological data was compiled, units were standardized where possible. The main issue
when compiling data was the lack of standardization of the calcification rates. A variety of
calcification rate units were used across different studies. Where possible, the units were converted
to mmol of $CaCO_3$ g weight$^{-1}$ hr$^{-1}$. However, the data required to do so was not always readily
available. Other units used for calcification rate were mmol of $CaCO_3$ m$^{-2}$ h$^{-1}$ and mmol of $CaCO_3$
m$^{-3}$ hr$^{-1}$, and there was also data used as an indication of calcification rate with units mmol #$^{-1}$ h$^{-1}$,
mmol h$^{-1}$, mmol cm$^{-2}$, % h$^{-1}$, where '#' indicates one individual. Growth rates and PIC production
rates were used as indicators of calcification rate for single-cell organisms. For some species, direct
calcification rates were not reported in the literature, instead only relevant parameters related to
calcification (shell length, density, thickness) over time were available from the experimental
studies. The decision was made to also collect these additional datasets because the statistical
analyses of this study focus on the trend in the absolute numbers and would not change by being
transformed into the rates. Data were analyzed on a species level, wherever rate units were the
same. Hereafter, this is referred to as the species rate group. Where there were multiple studies
available for the calcification rate of one species using the same rate units, the data were combined
(e.g. *Emiliania huxleyi*).

## 2.4 Sorting species-specific responses into categories per calcification response

Responses were split into 6 categories: linear positive and linear negative, parabolic, threshold
positive and negative, and neutral. The response was determined with a best-fit regression model,
using the ordinary least squares method in Statsmodels for Python (see Seabold et al., 2010). See
Fig. 2 for examples of these responses of calcification rate to increasing TA:DIC ratio.
The final response for each species was determined by the regression with the lowest p-value. This
method is in contrast with the Ries et al. (2009) study where they chose the regression analysis
that yielded the lowest square root of the mean squared error (RMSE) for a given species, and that
was statistically significant ($p \leq 0.05$). When applying their method to our data, parabolic and
exponential regressions were always favored over linear regressions. When examining these
regressions, we found that choosing the best fit based on the lowest p-value yielded better fits, as
this method prevents overfitting due to noise in the data. Where a linear regression had the best fit,
we assigned a linear response, which could be either positive or negative based on the slope. The
species with a significant exponential fit were categorized as threshold positive (+) or threshold
negative (-), which was distinguished from the parabolic response with the fitted parabolic curve.
The best fit regression was assigned to each species and plotted, but only if the p-value was
considered significant, i.e. lower than 0.05. These regressions were plotted along with a 90%
prediction interval, which accounts for the variability of the experimental data. The species with a
p-value > 0.05 were categorized as having no correlation (neutral response).
When multiple datasets obtained from different studies for the same species and rate units could
not be combined, we took each response into consideration and reported the p-value and RMSE
for each of the studies. Even when different studies reported varying calcification rates for the
same species, we refrained from selecting a single overall species response; rather, we analyzed
each species individually. The TA:DIC threshold was computed to indicate the point at which the
current calcification rate (i.e. the calcification rate at the baseline) is reduced by a half for linear
negative, threshold negative and parabolic responders. The thresholds and the amount of NaOH
and $Na_2CO_3$ required (starting at 10 µmol/kg and then in steps of 50 µmol/kg) to reach this
threshold were determined. For parabolic responders, the inflection points that tell us when the
rate is predicted to change slope are also included in Supplemental Table 2. Once the species'
responses were determined, an attempt was made to group them based on functional groups.
However, since species within the same functional group had varying responses, grouping them
together meant these responses were no longer visible due to a wide spread of data. Therefore,
most of the analysis remained on the species level (Table 1).

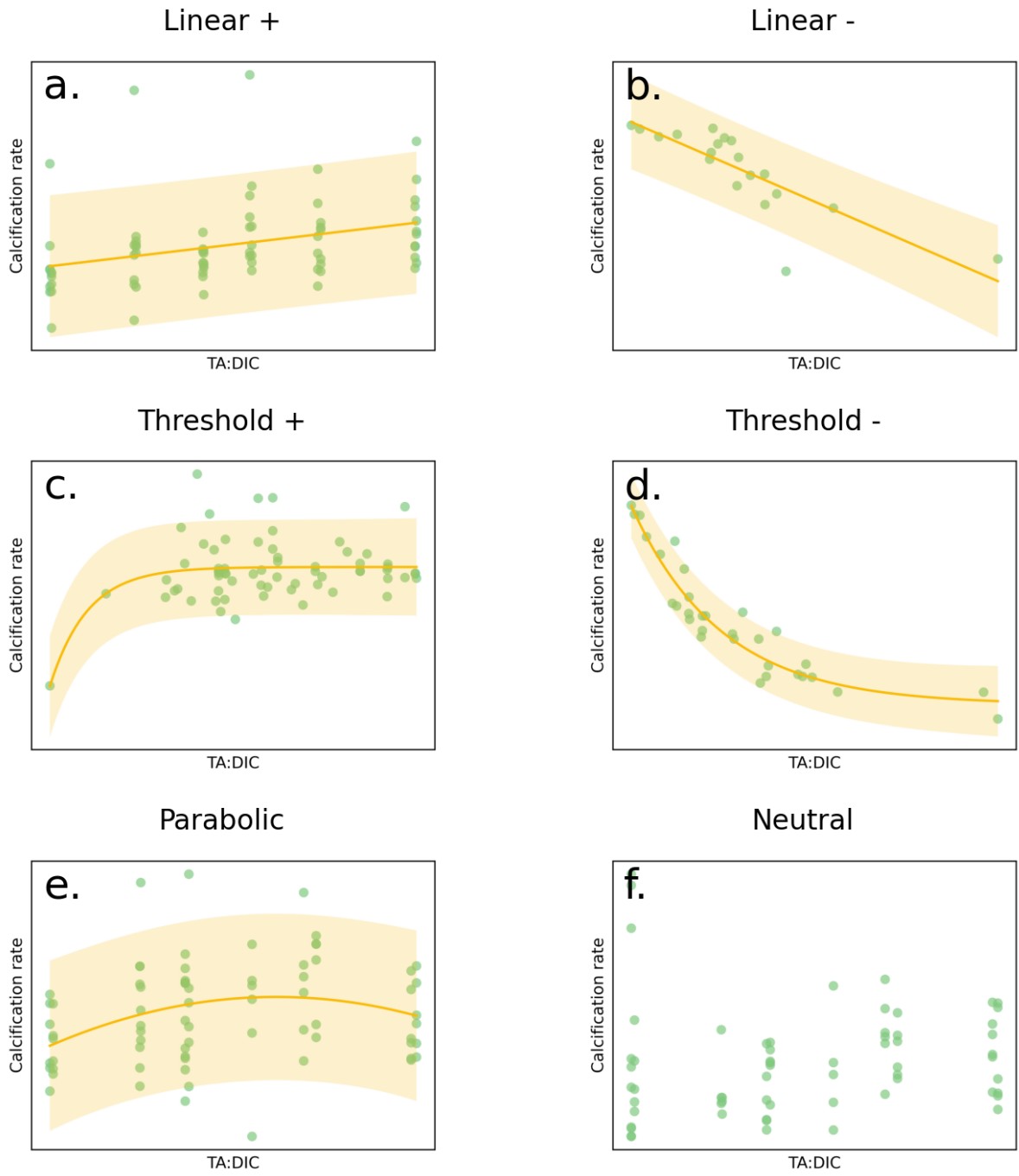


**Figure 2:** *Examples of the categories of responses between carbonate chemistry parameters*

*(TA:DIC) and calcification rate: a) linear positive (calcification increase with increased TA:DIC);*
*b) linear negative (calcification decrease with increased TA:DIC); c) exponential for the threshold*
*positive response (calcification increase, plateauing at higher TA:DIC); d) exponential for the*
*threshold negative response (calcification decline, plateauing at lower TA:DIC), e parabolic*
*(calcification increase followed by a decrease at higher TA:DIC) and f) neutral (non-significant)*
*response. Responses were only considered significant when p < 0.05, otherwise they were*
*categorized as neutral. Yellow shading represents the 90% prediction interval. Note that TA:DIC*
*on the x-axis corresponds to $pH_T$ and $\Omega_{ar}$, as these variables have an approximately linear*
*relationship at a particular salinity, temperature and pressure (see Fig. 1).*

## 2.5 Conceptual framework to evaluate increases in TA:DIC

The regression models applied to each species could be used to predict calcification rates at higher
TA:DIC ratio. We conceptually added alkalinity from the current calcification rate baseline. This
baseline was computed for each species using CO2SYS with $pCO_2$ = 425 ppm and $pH_T$ = 8.1, for
the average temperature and salinity for each species rate group, based on their respective OA
dataset(s) (see Supplemental Table 3). All CO2SYS calculations in this study were carried out
with the Python version of CO2SYS (Humphreys et al., 2022) using the stoichiometric dissociation
constants for carbonic acid from Sulpis et al. (2020), for sulfuric acid by Dickson et al. (1990) and
for total boron from Uppström (1974). From this baseline, TA was added in the form of both NaOH
and $Na_2CO_3$ to approximate changes in the carbonate chemistry settings, with NaOH changing
TA:DIC in the 1:1 ratio, and $Na_2CO_3$ inducing a 2:1 TA:DIC change. For example, 10 µmol/kg of
NaOH addition will increase TA by 10 µmol/kg and not affect DIC. For $Na_2CO_3$, 10 µmol/kg
addition will increase TA by 10 µmol/kg and increase DIC by 5 µmol/kg. Figure 1 demonstrates
the usefulness of this approach. For both NaOH and Na2CO3, 10 µmol/kg was conceptually added
using the principles of mass balance approach for the carbonate system via CO2SYS. This was
repeated for increments of 50 µmol/kg. We show this incremental addition in the plots up to a total
of 500 µmol/kg when generating the plots. When computing the thresholds, we added up to 1400
µmol/kg NaOH. The new TA:DIC ratios were estimated by adding the direct effect of $\Delta$TA and
$\Delta$DIC due to chemical additions of NaOH (assume $\Delta$DIC = 0) or Na2CO3 (assume $\Delta$DIC =
0.5*$\Delta$TA). A maximum of 500 µmol/kg was chosen to have more realistic additions of TA that
resemble those appropriate within the OAE field trials (e.g. Wang et al., 2023). With the new

TA:DIC ratios after TA addition, the species' regression models based on the fitted OA response
data were used to compute respective calcification rates (note that added points with NaOH or
Na2CO3 were not calculated as part of the regression). These data points were all plotted along
with the experimental data, regression model and prediction intervals as shown in Fig. 3.
We also determine the amount of NaOH and $Na_2CO_3$ needed to reach $pH_T$ 9 for each study.
This was computed for each species rate group using CO2SYS starting from $pCO_2 = 425$ ppm and
$pH_T = 8.1$, using the average temperature and salinity per species rate group, and by adding NaOH
or $Na_2CO_3$ in increments of 50 µmol/kg until $pH_T$ 9 was reached. Note that this method does not
incorporate gas exchange with the atmosphere, any biological processes, organic matter effects,
nitrification/denitrification, complexation, speciation or sediment-water interactions.

**2.6 Evaluation of the biological responses based on alkalinity addition**

The individual species with significant correlations were grouped visually based on their best-fit
regression models and are classified into positive, negative, and neutral as the following:
1) *Positive responders*: species with predicted *linear positive* and *threshold positive* calcification
rate response with increased TA addition.
2) *Negative responders*: species with predicted *linear negative*, *parabolic* and *threshold negative*
*response* in calcification rate upon (a certain amount of) TA addition. For the parabolic responders,
a concentration of NaOH was determined that indicates the threshold in TA:DIC beyond which
the response becomes negative (see inflection points in Supplemental Table 2).
3) *Neutral responders:* species with *no significant correlation* ($p < 0.05$) in calcification rate upon
TA addition.

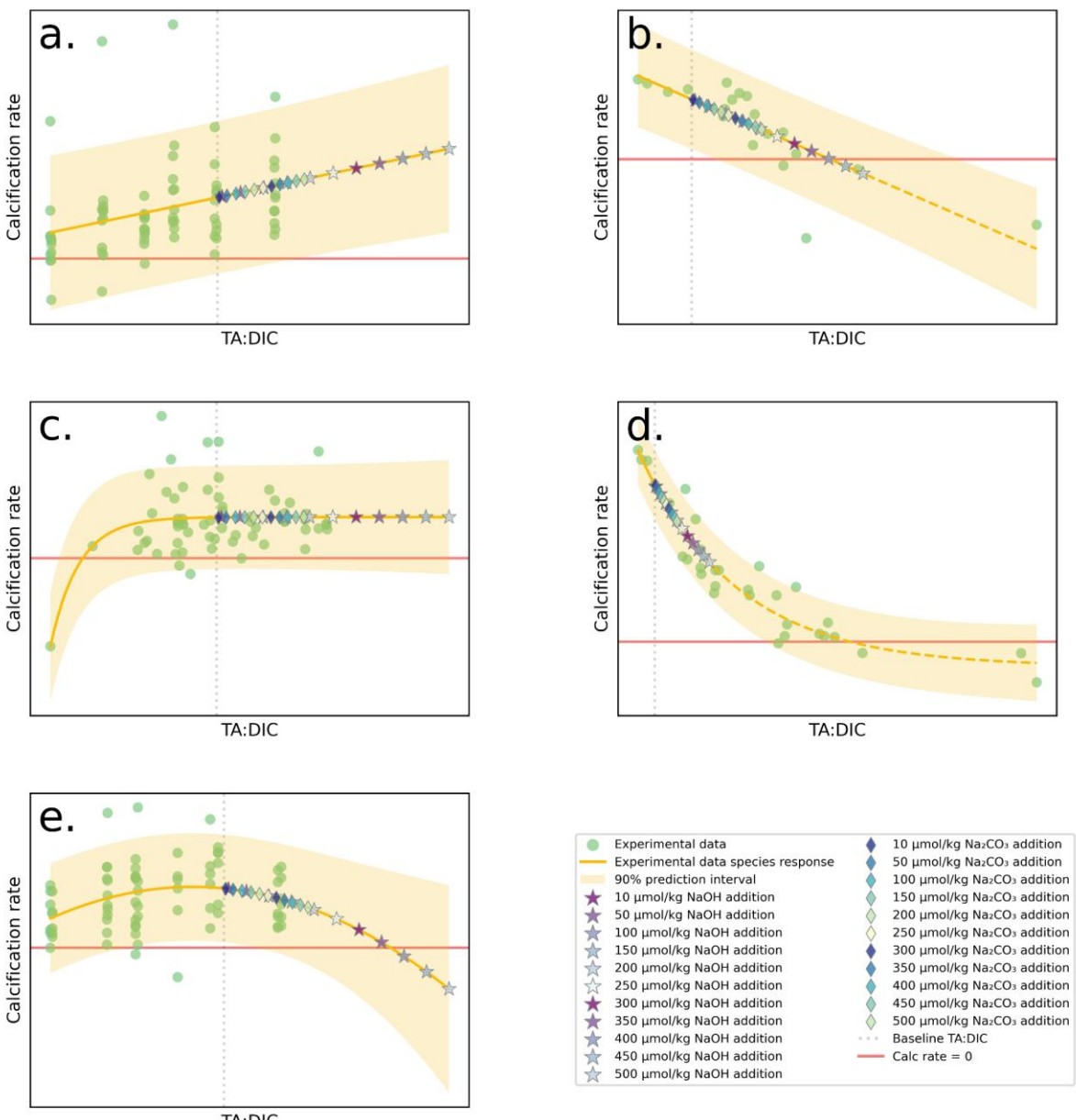


**Figure 3***: Conceptual diagrams for five types of responses; a) linear positive; b) linear negative;*
*c) threshold positive; d) threshold negative and e) parabolic response, plotted with experimental*
*data from OA studies (green dots), predicted values at various additions of alkalinity (stars and*
*diamonds), the regression line and prediction error margins fitted for a given species. The red*
*horizontal line indicates zero net dissolution (calcification rate is equal to 0; dissolution rate =*
*calcification rate). The grey vertical line indicates the baseline from which alkalinity is added.*
*NaOH and Na$_2$CO$_3$ addition is shown up to 500 μmol/kg.*

**2.7 Determining threshold values indicative of negative biological response to OAE**

The metrics to evaluate the sensitivity of calcification rate of the negative responders in this study were based on the amount of NaOH or $Na_2CO_3$ addition required to reduce the current calcification rate by a half. The greater the TA:DIC ratio value was required to trigger half calcification rate reduction, the less sensitive species was to NaOH addition. We refer to this TA:DIC ratio as the biological threshold, which we also report along with corresponding pH and $\Omega_{ar}$ and the associated uncertainty. TA:DIC thresholds were converted to their respective pH and $\Omega_{ar}$, which are affected by temperature and salinity. To calculate threshold pH and $\Omega_{ar}$ we used the average temperature and salinity per species rate group, as done for calculating the baseline.

**2.8 Extraction of the carbonate chemistry data from the GLODAP dataset**

We extracted total alkalinity, dissolved inorganic carbon, $\Omega_{ar}$, and $pH_T$ from the Global Ocean Data Analysis Project GLODAPv2.2023 dataset (https://glodap.info). We used the regression application in MATLAB with a second-order polynomial equation to predict $\Omega_{ar}$ from the TA:DIC. The regression analysis was performed using data from various depth intervals (0–10m, 0–30m, 0–50m, 0–100m, 0–200m) regionally and globally. The regional analysis divided the global oceans into the following groupings: Arctic (north of 65°N), Southern (south of 40°S), North Pacific (north of 40°N), Central Pacific (40°S to 40°N), North Atlantic (North of 40°N), Central Atlantic (40°S to 40°N), and Indian Ocean (north of 40°S).

**3. Results**

**3.1 Data collection for the calcification rate responses of different biological groups**

We examined 68 datasets, which covered 84 different species that were divided into 11 different groups (Fig. 4). These functional groups were corals (20% of datasets), calcifying algae (18%), mollusks (14%), foraminifera (10%), dinoflagellates (10%), coccolithophores (4%), gastropods (8%), crustaceans (5%), echinoderms (4%), pteropods (5%), and annelids (1%). In the mollusks group, we have separated out the gastropod and pteropod because of a higher number of studies that explicitly cover these two groups. The group of gastropods refers to all gastropods that are not pteropods. If all three groups were combined (mollusks, gastropods, pteropods), this group would be the largest.

364

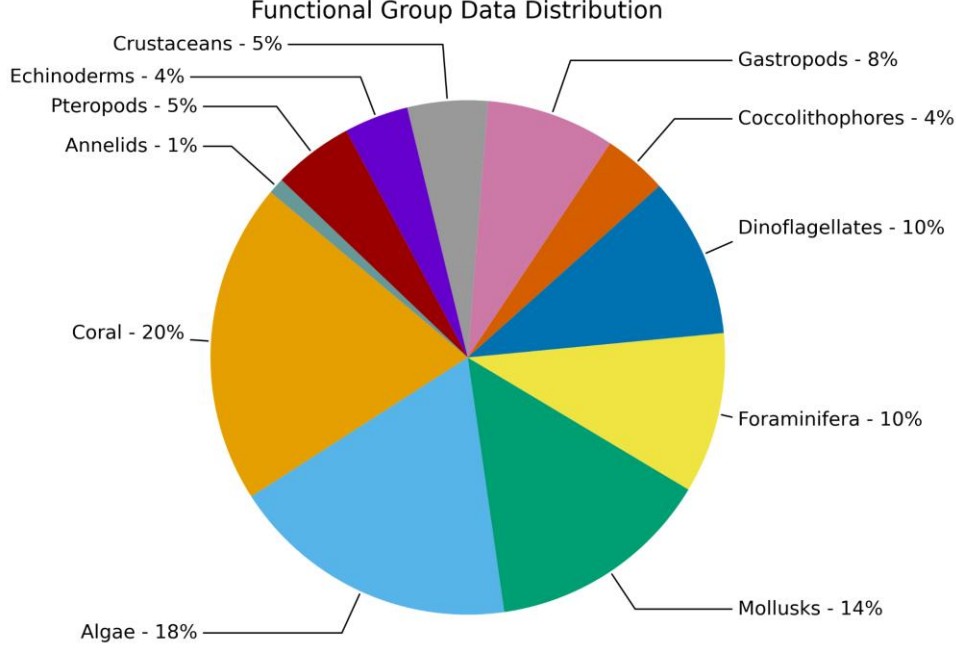

365

**Figure 4**: *Percent of studies for multiple groups (N=11) with available data for the calcification rate responses as part of data compilation of 68 studies covering 84 species).*

**3.2 Species-specific responses to NaOH/Na$_2$CO$_3$ addition**

Calcification rate responses of species from different groups were correlated to TA:DIC and summarized to obtain calcification rate response. The calcification rate responses encompassed linear (positive and negative), threshold (positive and negative), parabolic, and neutral responses, with the slope and the intercept of the response determining the type and the magnitude of the response. We present fitted responses of calcification rate per TA:DIC ratio for each examined species (Table 1; Supplemental Fig. 4). When possible, we fit a regression to multiple datasets of the same species that used the same calcification units. We also present the response with the additions of NaOH and Na$_2$CO$_3$ for each species per examined study and corresponding rate unit and their biological TA:DIC thresholds (Table 2; Supplemental Table 4).

**Table 1:** *The summary of all the OA studies from which the chemical and biological data was collected, including the name of the species and group and the accompanying calcification rate unit. The response for each species rate group was determined by the regression with the lowest*

 *p-value, where the p-value was smaller than 0.05. We also include the p-value, goodness of fit ($R^2$)*

 *and Root Mean Square Error (RMSE). Non-significant responses are categorized as having a*

 *'neutral' response. The type of response (linear positive or negative, threshold positive or*

 *negative, parabolic, and neutral) is indicated, as well as if this response is positive, negative or*

 *neutral.*

| Studies | n | Group | Species | Rate unit | Response | Pos/Neg/ Neut | p-value | $R^2$ | RMSE |
|---|---|---|---|---|---|---|---|---|---|
| Vasquez-Elizondo et al. (2016) | 4 | Algae | *Amphiroa tribulus* | mmol/m²/hr | neutral | Neutral | | | |
| Sinutok et al. (2011) | 16 | Algae | *Halimeda cylindracea* | mmol/hr | neutral | Neutral | | | |
| Comeau et al. (2013) | 71 | Algae | *Halimeda macroloba* | mmol/g/hr | parabolic | Negative | 0.0127 | 0.1200 | 0.0028 |
| Meyer et al. (2015) | 24 | Algae | *Halimeda macroloba* | mmol/m²/hr | neutral | Neutral | | | |
| Sinutok et al. (2011) | 16 | Algae | *Halimeda macroloba* | mmol/hr | parabolic | Negative | 0.0108 | 0.5000 | 0.0001 |
| Comeau et al. (2013) | 62 | Algae | *Halimeda minima* | mmol/g/hr | neutral | Neutral | | | |
| Meyer et al. (2015) | 24 | Algae | *Halimeda opuntia* | mmol/m²/hr | linear + | Positive | 0.0080 | 0.2800 | 0.0222 |
| Comeau et al. (2013) | 72 | Algae | *Hydrolithon reinboldii* | mmol/g/hr | linear + | Positive | 0.0053 | 0.1100 | 0.0026 |
| Cornwall et al. (2018) | 23 | Algae | *Hydrolithon reinboldii* | mmol/m²/hr | neutral | Neutral | | | |
| Comeau et al. (2013) | 72 | Algae | *Lithophyllum flavescens* | mmol/g/hr | neutral | Neutral | | | |
| Johnson et al. (2021) | 420 | Algae | *Lithophyllum sp.* | mmol/g/hr | linear + | Positive | 0.0000 | 0.1000 | 0.1136 |
| Vasquez-Elizondo et al. (2016) | 4 | Algae | *Lithothamnion sp.* | mmol/m²/hr | neutral | Neutral | | | |
| Monserrat et al. (2022) | 62 | Algae | *Neogoniolithon brassica-florida* | mmol/m²/hr | neutral | Neutral | | | |
| Ries et al. (2009) | 42 | Algae | *Neogoniolithon sp.* | mmol/g/hr | parabolic | Negative | 0.0000 | 0.4100 | 0.0003 |
| Vasquez-Elizondo et al. (2016), Comeau et al. (2018) | 26 | Algae | *Neogoniolithon sp.* | mmol/m²/hr | neutral | Neutral | | | |
| Briggs-Carpenter et al. (2019) | 425 | Algae | *Porolithon onkodes* | mmol/m²/hr | linear + | Positive | 0.0010 | 0.0300 | 0.8093 |
| Comeau et al. (2018, 2019) | 64 | Algae | *Sporolithon durum* | mmol/m²/hr | parabolic | Negative | 0.0012 | 0.2000 | 0.1704 |
| Ries et al. (2009) | 41 | Annelid | *Hydroides crucigera* | mmol/g/hr | neutral | Neutral | | | |
| Fiorini et al. (2011), Langer et al. (2006, 2011) | 14 | Cocco. | *Calcidiscus leptoporus* | mmol/#/hr | neutral | Neutral | | | |
| * | 233 | Cocco. | *Emiliania huxleyi* | mmol/#/hr | parabolic | Negative | 0.0000 | 0.1600 | 0.0000 |
| Casareto et al. (2009) | 14 | Cocco. | *Pleurochrysis carterae* | mmol/m³/hr | neutral | Neutral | | | |
| White et al. (2018) | 118 | Cocco. | *Pleurochrysis carterae* | mmol/# | neutral | Neutral | | | |
| Meyer et al. (2016) | 24 | Coral | *Acropora millepora* | mmol/m²/hr | neutral | Neutral | | | |
| Camp et al. (2017), Comeau et al. (2013) | 74 | Coral | *Acropora pulchra* | mmol/m²/hr | parabolic | Negative | 0.0000 | 0.2900 | 1.3257 |
| Agostini et al. (2021) | 18 | Coral | *Acropora solitaryensis* | mmol/m²/hr | neutral | Neutral | | | |
| Comeau et al. (2018), Comeau et al. (2019) | 81 | Coral | *Acropora yongei* | mmol/m²/hr | linear + | Positive | 0.0000 | 0.2900 | 1.9447 |
| Bove et al. (2020) | 27 | Coral | *Duncanopsammia axifuga* | mmol/m²/hr | linear + | Positive | 0.0016 | 0.3300 | 5.0785 |
| Cornwall et al. (2018) | 44 | Coral | *Goniopora sp.* | mmol/m²/hr | neutral | Neutral | | | |
| Maier et al. (2009) | 237 | Coral | *Lophelia pertusa* | mmol/g/hr | linear + | Positive | 0.0030 | 0.0400 | 0.0002 |
| Bove et al. (2020) | 65 | Coral | *Montastraea cavernosa* | mmol/m²/hr | linear + | Positive | 0.0154 | 0.0900 | 0.5047 |
| Ries et al. (2009) | 54 | Coral | *Oculina arbuscula* | mmol/g/hr | parabolic | Negative | 0.0000 | 0.8600 | 0.0001 |
| Comeau et al. (2013) | 72 | Coral | *Pavona cactus* | mmol/m²/hr | parabolic | Negative | 0.0002 | 0.2200 | 0.9093 |
| Comeau et al. (2019) | 49 | Coral | *Plesiastrea versipora* | mmol/m²/hr | linear + | Positive | 0.0069 | 0.1500 | 0.6003 |
| Brown et al. (2022) | 4 | Coral | *Pocillopora damicornis* | mmol/g/hr | neutral | Neutral | | | |
| Comeau et al. (2013, 2018), Putnam-Gates et al. (2015) | 117 | Coral | *Pocillopora damicornis* | mmol/m²/hr | neutral | Neutral | | | |

| Studies | n | Group | Species | Rate unit | Response | Pos/Neg/ Neut | p-value | $R^2$ | RMSE |
|---|---|---|---|---|---|---|---|---|---|
| Evensen-Edmunds et al. (2016) | 60 | Coral | *Pocillopora verrucosa* | mmol/m²/hr | linear + | Positive | 0.0132 | 0.1000 | 0.8297 |
| Agostini et al. (2021) | 18 | Coral | *Porites heronensis* | mmol/m²/hr | neutral | Neutral | | | |
| Comeau et al. (2013) | 72 | Coral | *Porites rus* | mmol/m²/hr | linear + | Positive | 0.0020 | 0.1300 | 2.0281 |
| Okazaki et al. (2013) | 75 | Coral | *Siderastrea radians* | mmol/m²/hr | linear + | Positive | 0.0004 | 0.1600 | 2.7886 |
| Okazaki et al. (2013) | 64 | Coral | *Solenastrea hyades* | mmol/m²/hr | threshold + | Positive | 0.0004 | 0.2300 | 2.0385 |
| Krueger et al. (2017) | 36 | Coral | *Stylophora pistillata* | mmol/m²/hr | neutral | Neutral | | | |
| Pansch et al. (2014) | 36 | Crust. | *Amphibalanus improvisus* | mmol/g/hr | linear + | Positive | 0.0000 | 0.4300 | 0.0004 |
| Ries et al. (2009) | 36 | Crust. | *Callinectes sapidus* | mmol/g/hr | linear - | Negative | 0.0000 | 0.4000 | 0.0082 |
| Ries et al. (2009) | 18 | Crust. | *Homarus americanus* | mmol/g/hr | linear - | Negative | 0.0014 | 0.4800 | 0.0079 |
| Ries et al. (2009) | 12 | Crust. | *Penaeus plebejus* | mmol/g/hr | linear - | Negative | 0.0124 | 0.4800 | 0.0006 |
| Findlay et al. (2010) | 6 | Crust. | *Semibalanus balanoides* | mmol/g/hr | neutral | Neutral | | | |
| Tatters et al. (2013) | 45 | Dino. | *Alexandrium sp.* | 1/hr | neutral | Neutral | | | |
| Hansen et al. (2007) | 19 | Dino. | *Ceratium lineatum* | #/hr | linear - | Negative | 0.0000 | 0.6700 | 0.0043 |
| Tatters et al. (2013) | 45 | Dino. | *Gonyaulax sp.* | 1/hr | neutral | Neutral | | | |
| Hansen et al. (2007) | 31 | Dino. | *Heterocapsa triquetra* | #/hr | threshold - | Negative | 0.0000 | 0.9100 | 0.0027 |
| Wang et al. (2019) | 4 | Dino. | *Karenia mikimotoi* | 1/hr | neutral | Neutral | | | |
| Tatters et al. (2013) | 45 | Dino. | *Lingulodinium polyedrum* | 1/hr | neutral | Neutral | | | |
| Tatters et al. (2013) | 45 | Dino. | *Prorocentrum micans* | 1/hr | neutral | Neutral | | | |
| Hansen et al. (2007) | 21 | Dino. | *Prorocentrum minimum* | #/hr | threshold - | Negative | 0.0000 | 0.8800 | 0.0019 |
| Brading et al. (2011) | 175 | Dino. | *Symbiodinium sp.* | #/hr | linear - | Negative | 0.0010 | 0.0600 | 0.0066 |
| Van de Waal et al. (2013) | 12 | Dino. | *Thoracosphaera heimii* | mmol/hr | parabolic | Negative | 0.0002 | 0.8500 | 0.0000 |
| Ries et al. (2009) | 17 | Echino. | *Arbacia punctulata* | mmol/g/hr | parabolic | Negative | 0.0000 | 0.8900 | 0.0003 |
| Courtney et al. (2013) | 4 | Echino. | *Echinometra viridis* | %/hr | linear + | Positive | 0.0244 | 0.9500 | 2.3854 |
| Courtney et al. (2015) | 28 | Echino. | *Echinometra viridis* | % | linear + | Positive | 0.0009 | 0.3500 | 13.0388 |
| Ries et al. (2009) | 18 | Echino. | *Eucidaris tribuloides* | mmol/g/hr | threshold + | Positive | 0.0000 | 0.8400 | 0.0004 |
| Keul et al. (2013) | 205 | Foram. | *Ammonia sp.* | mmol/#/hr | linear - | Negative | 0.0277 | 0.0200 | 0.0000 |
| Prazeres et al. (2015) | 32 | Foram. | *Amphistegina lessonii* | %/hr | parabolic | Negative | 0.0008 | 0.3900 | 0.0010 |
| Kisakurek et al. (2011) | 16 | Foram. | *Globigerinella siphonifera* | mmol/hr | neutral | Neutral | | | |
| Kisakurek et al. (2011) | 14 | Foram. | *Globigerinoides ruber* | mmol/#/hr | neutral | Neutral | | | |
| Reymond et al. (2013) | 179 | Foram. | *Marginopora rossi* | %/hr | linear + | Positive | 0.0000 | 0.1900 | 0.0090 |
| Uthicke-Fabricius et al. (2012) | 47 | Foram. | *Marginopora vertebralis* | mmol/g/hr | threshold + | Positive | 0.0000 | 0.4000 | 0.0004 |
| Sinutok et al. (2011) | 16 | Foram. | *Marginopora vertebralis* | mmol/hr | neutral | Neutral | | | |
| Prazeres et al. (2015) | 32 | Foram. | *Marginopora vertebralis* | %/hr | linear - | Negative | 0.0006 | 0.3300 | 0.0005 |
| Manno et al. (2012) | 192 | Foram. | *Neogloboquadrina pachyderma* | mmol/#/hr | linear + | Positive | 0.0000 | 0.7100 | 0.0000 |
| Oron et al. (2020) | 96 | Foram. | *Operculina ammonoides* | mmol/g/hr | linear - | Negative | 0.0031 | 0.0900 | 0.0017 |
| Manriquez et al. (2016) | 74 | Gastropod | *Concholepas concholepas* | mmol/g/hr | linear + | Positive | 0.0000 | 0.2400 | 0.0009 |
| Noisette et al. (2016), Ries et al. (2009) | 173 | Gastropod | *Crepidula fornicata* | mmol/g/hr | parabolic | Negative | 0.0000 | 0.2100 | 0.0028 |
| Garilli et al. (2015) | 68 | Gastropod | *Cyclope neritea* | mmol/g/hr | linear - | Negative | 0.0020 | 0.1400 | 0.0037 |
| Ries et al. (2009) | 42 | Gastropod | *Littorina littorea* | mmol/g/hr | linear + | Positive | 0.0001 | 0.3400 | 0.0002 |
| Bibby et al. (2007) | 4 | Gastropod | *Littorina littorea* | µm (shell thickness) | neutral | Neutral | | | |
| Garilli et al. (2015) | 315 | Gastropod | *Nassarius corniculus* | mmol/g/hr | parabolic | Negative | 0.0000 | 0.2500 | 0.0064 |
| Ries et al. (2009) | 21 | Gastropod | *Strombus alatus* | mmol/g/hr | linear + | Positive | 0.0000 | 0.6400 | 0.0001 |
| Ries et al. (2009) | 33 | Gastropod | *Urosalpinx cinerea* | mmol/g/hr | linear + | Positive | 0.0000 | 0.5700 | 0.0001 |
| Ries et al. (2009) | 18 | Mollusks | *Argopecten irradians* | mmol/g/hr | linear + | Positive | 0.0097 | 0.3500 | 0.0002 |
| Ramajo et al. (2016) | 6 | Mollusks | *Argopecten purpuratus* | mmol/g/hr | neutral | Neutral | | | |
| Zhang et al. (2011) | 5 | Mollusks | *Azumapecten farreri* | mmol/g/hr | linear + | Positive | 0.0106 | 0.9200 | 0.0001 |
| Ong et al. (2017) | 24 | Mollusks | *Cerastoderma edule* | mmol/g/hr | neutral | Neutral | | | |

| Studies | n | Group | Species | Rate unit | Response | Pos/Neg/Neut | p-value | R² | RMSE |
|---|---|---|---|---|---|---|---|---|---|
| Sordo et al. (2021) | 27 | Mollusks | *Chamelea gallina* | mmol/g/hr | neutral | Neutral | | | |
| Gazeau et al. (2007) | 20 | Mollusks | *Crassostrea gigas* | mmol/g/hr | linear + | Positive | 0.0001 | 0.6100 | 0.0000 |
| Ries et al. (2009), Waldbusser et al. (2011) | 28 | Mollusks | *Crassostrea virginica* | mmol/g/hr | threshold + | Positive | 0.0000 | 0.5600 | 0.0003 |
| Ries et al. (2009) | 25 | Mollusks | *Mercenaria mercenaria* | mmol/g/hr | threshold + | Positive | 0.0000 | 0.8300 | 0.0000 |
| Ries et al. (2009) | 14 | Mollusks | *Mya arenaria* | mmol/g/hr | linear + | Positive | 0.0001 | 0.7300 | 0.0003 |
| Ninokawa et al. (2020) | 13 | Mollusks | *Mytilus californianus* | mmol/m²/hr | neutral | Neutral | | | |
| Ries et al. (2009), Gazeau et al. (2007) | 86 | Mollusks | *Mytilus edulis* | mmol/g/hr | linear + | Positive | 0.0119 | 0.0700 | 0.0002 |
| Gazeau et al. (2014) | 11 | Mollusks | *Mytilus galloprovincialis* | mmol/g/hr | neutral | Neutral | | | |
| Cameron et al. (2019) | 30 | Mollusks | *Pecten maximus* | mmol/g/hr | neutral | Neutral | | | |
| Comeau et al. (2010b) | 5 | Pteropod | *Cavolinia inflexa* | mm (shell length) | neutral | Neutral | | | |
| Comeau et al. (2009, 2010a) | 12 | Pteropod | *Limacina helicina* | mmol/g/hr | linear + | Positive | 0.0000 | 0.8500 | 0.0001 |
| Lischka et al. (2011, 2012) | 119 | Pteropod | *Limacina helicina* | mm (shell length) | threshold + | Positive | 0.0003 | 0.1300 | 0.1303 |
| Bednarsek (2021a), Mekkes et al. (2021) | 117 | Pteropod | *Limacina helicina* | µm (shell thickness) | parabolic | Negative | 0.0000 | 0.1800 | 0.0038 |
| Lischka et al. (2012) | 28 | Pteropod | *Limacina retroversa* | mm (shell length) | neutral | Neutral | | | |

*Barcelos-Ramos et al. (2010), Fiorini et al. (2011), Iglesias-Rodriguez et al. (2008), Richier et al. (2011), Sciandra et al. (2003), Stoll et al. (2012),
Gafar et al. (2018), Bach et al. (2011), Sett et al. (2014).
Within each of the 11 functional groups, several categories of calcification response occur within
each functional group, with the most varied being the group of dinoflagellates and foraminifera,
both showing 4 or 5 different categories of calcification responses (Fig. 5). Of the six types of
responses of calcification rate vs. TA:DIC, 28% were linear positive (N=27), 9% linear negative
(N=9), 6% threshold positive (N=6), 2% threshold negative (N=2), 15% parabolic (N=14) and
40% neutral (N=38).
Such responses could be further summed up into positive (linear and threshold positive), negative
(linear and threshold negative, parabolic) and neutral responses (Fig. 6) when generalized for
calcification rate against TA:DIC ratio. A summary of responses includes 34.4% positive (N=33),
26.0% negative (N=25), while 39.6% show a neutral response (N=38).

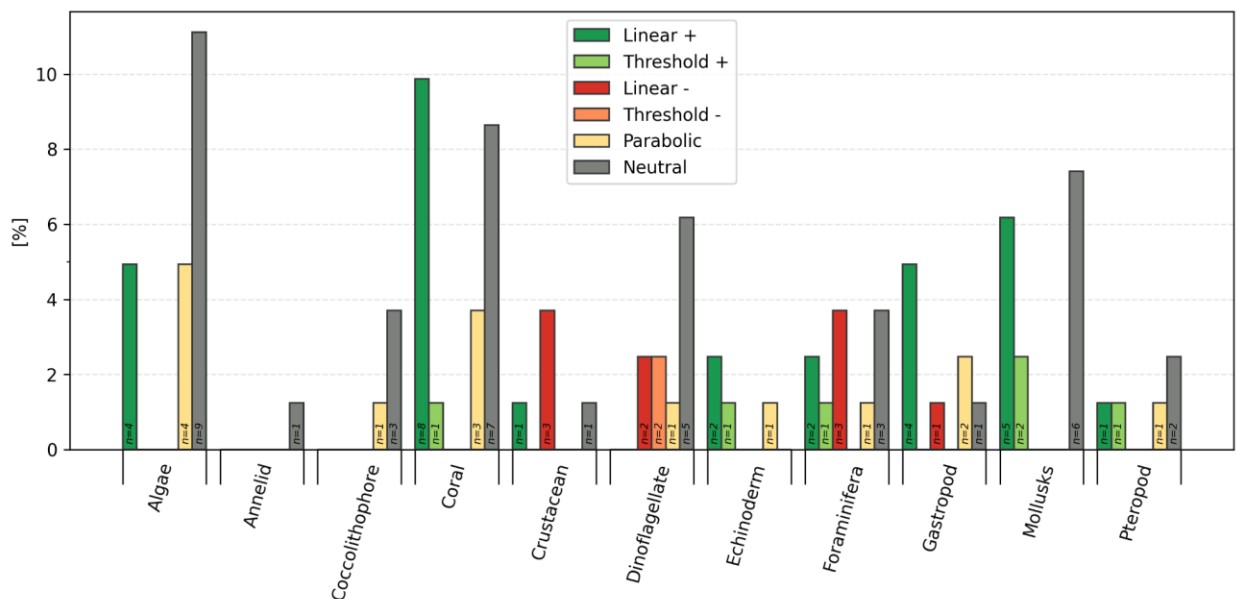


**Figure 5:** *Categories of calcification rate responses and percentage (%) response across eleven groups (calcifying algae, annelids, coccolithophores, corals, crustaceans, dinoflagellate, echinoderms, foraminifera, gastropods, mollusks, pteropods). The number on the bar indicates the number of studies of species included.*


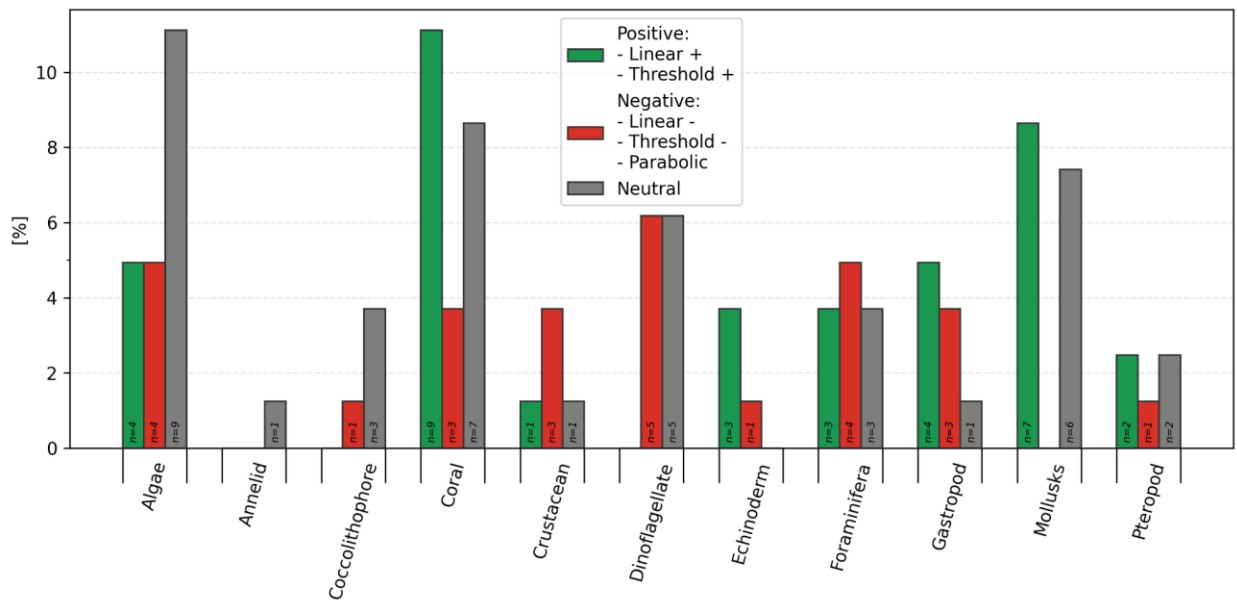


**Figure 6**: *Summary of percentage (%) responses in calcification rates as positive (linear and threshold positive), negative (linear and threshold negative, parabolic) and neutral across eleven groups (calcifying algae, annelids, coccolithophores, corals, crustaceans, dinoflagellate,*

*echinoderms, foraminifera, gastropods, mollusks, pteropods). The number on the bar indicates*
*the number of studies with species included.*

**3.3 Evaluation of the responses to NaOH/Na$_2$CO$_3$ addition**

Upon added TA, the calcification rate in positive responders will increase, either in a linear or
threshold positive response, where calcification plateaus, with the concentration being dependent
on the species-specific rate of response (Fig. 2; Supplemental Fig. 4). The negative responders
(linear or threshold negative and parabolic) will be negatively impacted as follows: first, for the
linear negative responders, addition of the Na$_2$CO$_3$ will linearly decrease calcification rate, but
there is no associated threshold to it; second, for the threshold negative responders, calcification
rate will decline in an exponential way until reaching a TA:DIC value where the response plateaus;
and third, for the parabolic responders, the calcification rate will initially increase until reaching a
certain TA:DIC threshold upon which calcification starts declining. The TA:DIC thresholds for
negative responders are species-specific (Table 2; Supplemental Table 4).

**3.4 Threshold values indicative of negative biological response to OAE**

The TA:DIC biological thresholds in Table 2 are determined by the amount of NaOH addition
required to reduce calcification rate by a half (see Supplemental Table 4 for Na$_2$CO$_3$ thresholds).
These thresholds demonstrate the range of carbonate chemistry conditions over which the negative
biological effects of OAE deployment might occur and are shown alongside the corresponding
pH$_T$ and $\Omega_{ar}$. Uncertainties are higher for the experimental studies where the experimental
temperature and salinity ranges were high (see Supplemental Table 5), seeing as we use the average
for each species rate group to compute the baseline and thresholds.
For the negative responders, TA:DIC thresholds range from 1.13 to 1.74. The majority of species
have reached their thresholds by an addition of 500 µmol/kg NaOH, though for 3 species a NaOH
addition of more than 500 µmol/kg is required to cross the thresholds in the TA:DIC range of 1.39
to 1.74. *Crepidula fornicata* (gastropod), *Neogoniolithon sp.* (algae), *Homarus americanus*
(crustacean) and *Oculina arbuscula* (coral) reach their thresholds by 100 µmol/kg addition of
NaOH, indicating they are more sensitive to alkalinity addition. Foraminifera, dinoflagellates and

coccolithophores generally require higher concentrations of NaOH to reach their thresholds, with the linear negative responder *Ammonia sp.* of the foraminifera group requiring 1400 µmol/kg to reduce calcification rate in half.

For some negative responders (*Arbacia punctulata, Nassarius corniculus, Penaeus plebejus, Callinectes sapidus, Cyclope neritea,* and *Symbiodinium sp*.), the baseline from which NaOH addition occurs was outside of the range of the experimental data and very close to a calcification rate of 0. These were omitted from Table 2 since our defined threshold does not give an accurate representation of their sensitivity to alkalinity addition. *Limacina helicina* was also omitted since the indicator of calcification (shell thickness) was not an actual rate.

**Table 2:** *Studies with negative responders (linear and threshold negative, parabolic) with demonstrated TA:DIC thresholds, indicating the amount of NaOH needed to halve the current calcification rate (i.e. at the baseline). The value for TA:DIC threshold is used to determine the $pH_T$ and $\Omega_{ar}$ (at average temperature and average salinity per species). See Supplemental Table 4 for $Na_2CO_3$ thresholds.*

| Studies | Group | Species | Temp (°C) | Salinity | Rate unit | Threshold | TA addition | pH$_T$ at threshold | ΔpH$_T$ from baseline | Ω$_{ar}$ at threshold | Exposure time |
|---|---|---|---|---|---|---|---|---|---|---|---|
| Noisette et al. (2016), Ries et al. (2009) | Gastropod | *Crepidula fornicata* | 15.31 | 34.33 | mmol/g/hr | 1.13 | 50 | 8.17 | 0.07 | 3.77 | 6 months / 60 days |
| Ries et al. (2009) | Algae | *Neogoniolithon sp.* | 25.00 | 31.70 | mmol/g/hr | 1.17 | 50 | 8.16 | 0.06 | 4.87 | 60 days |
| Ries et al. (2009) | Crustacean | *Homarus americanus* | 25.02 | 31.96 | mmol/g/hr | 1.19 | 100 | 8.22 | 0.12 | 5.49 | 60 days |
| Ries et al. (2009) | Coral | *Oculina arbuscula* | 25.01 | 31.61 | mmol/g/hr | 1.19 | 100 | 8.22 | 0.12 | 5.46 | 60 days |
| Prazeres et al. (2015) | Foraminifera | *Amphistegina lessonii* | 24.18 | 33.46 | %/hr | 1.21 | 150 | 8.27 | 0.17 | 6.10 | 30 days |
| Hansen et al. (2007) | Dinoflagellate | *Ceratium lineatum* | 15.00 | 30.00 | #/hr | 1.18 | 200 | 8.38 | 0.28 | 5.15 | 14 d acclimation; 7 days; 14 days exposure; 22 days stationary growth phase |
| Sinutok et al. (2011) | Algae | *Halimeda macroloba* | 27.23 | 36.27 | mmol/g/hr | 1.26 | 200 | 8.30 | 0.20 | 7.38 | 2 weeks acclimation, 2 weeks incubation |
| Comeau et al. (2019) | Algae | *Sporolithon durum* | 20.60 | 35.87 | mmol/m²/hr | 1.22 | 200 | 8.32 | 0.22 | 6.31 | 27 weeks |
| Van de Waal et al. (2013) | Dinoflagellate | *Thoracosphaera heimii* | 15.00 | 34.00 | mmol/hr | 1.23 | 300 | 8.46 | 0.36 | 6.56 | 21 days acclimation, 8 days experiment = total of >10 generations |
| Oron et al. (2020) | Foraminifera | *Operculina ammonoides* | 25.00 | 37.00 | mmol/g/hr | 1.33 | 400 | 8.46 | 0.36 | 9.44 | 65 - 120 hours |
| Prazeres et al. (2015) | Foraminifera | *Marginopora vertebralis* | 24.18 | 33.46 | %/hr | 1.33 | 450 | 8.53 | 0.43 | 9.78 | 30 days |
| Camp et al. (2017), Comeau et al. (2013) | Coral | *Acropora pulchra* | 27.30 | 36.27 | mmol/m²/hr | 1.38 | 500 | 8.52 | 0.42 | 11.05 | N7A (natural conditions) 2 weeks acclimation; 2 weeks incubation |
| Hansen et al. (2007) | Dinoflagellate | *Heterocapsa triquetra* | 15.00 | 30.00 | #/hr | 1.30 | 500 | 8.66 | 0.56 | 8.81 | 14 d acclimation; 7 days acclimation to experimental conditions; 14 days exposure; 22 days stationary growth phase |
| Comeau et al. (2013) | Coral | *Pavona cactus* | 27.23 | 36.28 | mmol/m²/hr | 1.38 | 500 | 8.52 | 0.42 | 11.03 | 2 weeks acclimation; 2 weeks incubation |
| Hansen et al. (2007) | Dinoflagellate | *Prorocentrum minimum* | 15.00 | 30.00 | #/hr | 1.39 | 700 | 8.81 | 0.71 | 11.35 | 14 d acclimation; 7 days acclimation to experimental conditions; 14 days exposure ; 22 days stationary growth phase |
| * | Coccolithophore | *Emiliania huxleyi* | 17.30 | 35.12 | mmol/#/hr | 1.46 | 850 | 8.83 | 0.73 | 13.65 | ** |
| Keul et al. (2013) | Foraminifera | *Ammonia sp.* | 26.00 | 32.75 | mmol/#/hr | 1.74 | 1400 | 9.11 | 1.01 | 22.27 | 59-96 days of culturing |

*Barcelos-Ramos et al. (2010), Fiorini et al. (2011), Iglesias-Rodriguez et al. (2008), Richier et al. (2011), Sciandra et al. (2003), Stoll et al. (2012), Gafar et al. (2018), Bach et al. (2011), Sett et al. (2014).

**26hrs, Acclimation for 7 generations, experiment/sampling for 2-3 generations, n/a, 8 days, 16 days, Acclimation for 12 generations, Pre-acclimation for 8-12 generations, 9 generations, Acclimated for at ~7 generations (5-15 days)

## 3.5 Regulatory pH$_T$ 9 threshold

We also compute how much NaOH and Na$_2$CO$_3$ needs to be added before reaching a pH$_T$ threshold of 9, as per the US Environmental Protection Agency's rule for waste water not exceeding a pH$_T$ of 9 when entering the coastal ocean (NPDES manual, 2010). This amount averages at 1200 µmol/kg of NaOH and 4700 µmol/kg of Na$_2$CO$_3$ for most of the examined species. For some species (*Amphibalanus improvisus, Neogloboquadrina pachyderma, Limacina helicina, Limacina retroversa, Lophelia pertusa,* and *Semibalanus balanoides*), their threshold was reached below 1000 µmol/kg NaOH and 3000 µmol/kg Na2CO3, with *Amphibalanus improvisus* reaching a

threshold at 750 µmol/kg NaOH and 2250 µmol/kg $Na_2CO_3$.
**3.6 Global and regional carbonate chemistry data coverage based on GLODAP datasets**
The compilation of chemical observational data (pH, $\Omega_{ar}$, TA, DIC) was done for the GLODAP
data across the regional ocean and global scales to determine the range of $\Omega_{ar}$, TA and DIC (as
represented by the TA:DIC ratio) and TA:DIC vs $\Omega_{ar}$ correlation down to the depths averaged over
200 m. This allowed us to apply the thresholds even for the regions for which we do not have
sufficient or reliable data or experimental coverage, making the inferences about the OAE impact
even in those regions.
Here, we focused on showing the results ranging over the 0–50m because this covers most of the
biological habitat for examined species and it is where the OAE enhancement would induce the
greatest changes. Over the 0–50 m depth, $\Omega_{ar}$ ranges from 0.2 to 5 and TA:DIC ranges from 0.1 to
1.25 and both parameters are correlated across all the regions, as demonstrated by the fitted second-
order polynomial regressions, with $R^2$ of 0.96 or higher, and all the correlations being significant
(Fig. 7), with regional specific relationships not impacting the fit. All the correlation parameters
are presented in Supplemental Table 4. Similar fits were found at different depths. The conditions
in the higher latitude regions are located at the lower range of $\Omega_{ar}$ vs TA:DIC, while the conditions
in the low latitudes and temperate regions are at the upper range, with the highest values present
in the central Atlantic and Pacific region. Such strong correlation as observed for $\Omega_{ar}$ vs TA:DIC
does not exist with pH, regardless of the depth interval examined. While the correlations are still
significant, they are broadly distributed and represented over a shorter TA:DIC range, with
significantly lower goodness of fit (Supplemental Fig. 4), with the correlations being highly
regionally dependent due to pH and temperature co-linearity. Because of this, all further biological
analyses are only done using the $\Omega_{ar}$ vs TA:DIC ratio.

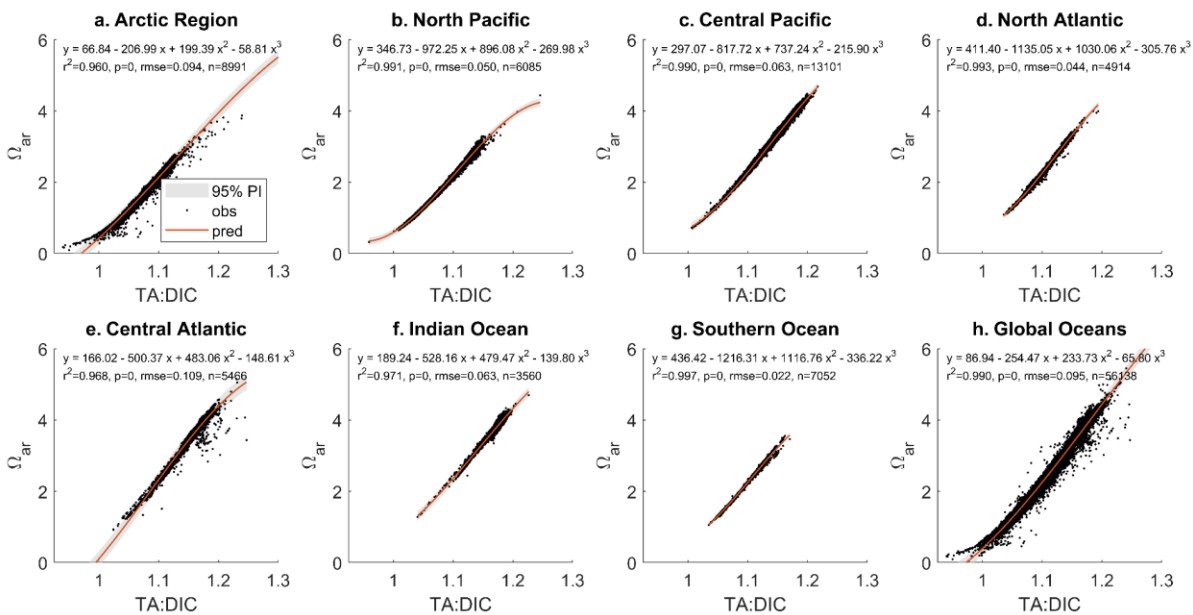


**Figure 7:** *The range of observed $\Omega_{ar}$, TA and DIC values (as represented by the TA:DIC ratio) values and the relationship with the best fitted curve between $\Omega_{ar}$ vs TA:DIC across regional (a-g) and global (h) scales based on the observational GLODAP dataset averaged over the 0-50 m depth range.*

### 3.7 TA:DIC vs $\Omega_{ar}$ for experimental data and GLODAP

We compared the ranges of TA:DIC and $\Omega_{ar}$ of biological experimental data with field biogeochemical data (GLODAP) to examine if similar range of conditions and TA:DIC correlations are applicable over a broader, global dataset. For this, we plotted $\Omega_{ar}$ vs TA:DIC along with the GLODAP regression line for $\Omega_{ar}$ vs TA:DIC (Fig. 8). For each TA and DIC datapoint, the corresponding salinity and temperature specific values for that data point were used to compute $\Omega_{ar}$. We show the similarity in the conditions, which gives the validity of our experimentally derived thresholds to be extrapolated within the global GLODAP dataset.

Figure 8 also shows that various biological groups are clustered around specific TA:DIC ratios, for example, mollusks, coral and coccolithophores are represented on the lower, mid, and higher TA:DIC spectra, respectively, while dinoflagellates are randomly scattered off the TA:DIC line. This indicates that there is a general lack of data distribution in the upper ranges of TA:DIC ratio, especially for the groups that are lying at the lower and mid end of the TA:DIC ratio spectra.

Plotting biological data from the OA datasets against the regional and global TA:DIC gradient
derived from GLODAP (Fig. 7), we also observed that experimental data ranges were not always
consistent with natural conditions, for example, having a lower $\Omega_{ar}$ at a higher TA:DIC ratio.

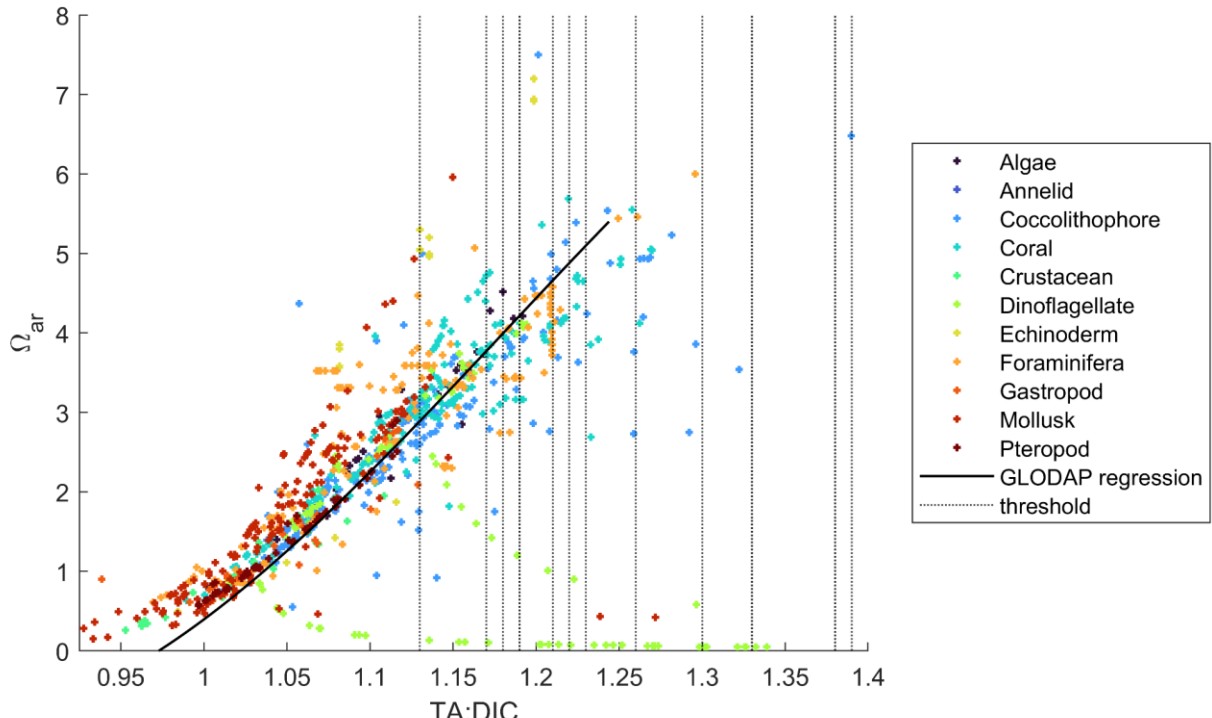


**Figure 8:** *$\Omega_{ar}$ values from experimental biological studies for eleven investigated functional*
*groups (see legend) plotted against TA:DIC, with the latter being computed using experimental*
*TA and DIC. The black line represents the regression line of TA:DIC and $\Omega_{ar}$ data from the*
*GLODAP dataset (covering 0-50m depth). See Supplemental Fig. 5 for GLODAP $\Omega_{ar}$ vs TA:DIC,*
*from which the black regression line shown here is derived. The vertical dotted lines represent*
*the thresholds shown in Table 2.*
**4. Discussion**
OAE is a quickly developing strategy that is in the field-testing phase despite extremely limited
understanding of the sequestration potential, biological implications and environmental concerns.
Hence, gaining insights of potential risks for the biological species and communities is essential
and timely. In retrospect, it took decades for the OA research community to get a more accurate
and comprehensive understanding leading to predictions of biological responses to OA (Riebesell
and Gattuso, 2015). Without a very clear conceptual strategy for the OAE testing, the research
community might also need years to decades before OAE-related implications are
comprehensively understood. Consequently, there is an essential need to develop an assessment
framework of predictive responses and testing strategies that will assist in OAE scaling and risk
avoidance. This paper aims at developing such an assessment, where calcification responses
against TA:DIC are categorized per species. We propose to use the TA:DIC ratio in the biological
studies reporting OAE results, as we believe it simplifies the system and makes it easier to use and
translate the carbonate chemistry in the experimental setting. Such a TA:DIC ratio allows to
ultimately standardize the biogeochemical and biological data and is useful for easier comparisons
among the conducted experiments.
**4.1. Identified strengths and limitations of the synthesis approach based on OA studies**
Prior to conducting this study, several drawbacks were identified that could potentially limit such
a synthesis work: first, an insufficient amount of data at the upper range of carbonate chemistry
conditions (high pH, high $\Omega_{ar}$); second, experimental data under conditions with no relevance to
natural settings (Fig. 8); and third, an insufficient number of validation studies under high TA
conditions to validate the results of this synthesis. To overcome the first two limitations, the
decision was made to combine multiple OA datasets for a single species with the aim to achieve a
greater range in carbonate chemistry conditions, including higher pH, $\Omega_{ar}$ experimental values,
which should reduce the uncertainty of the predictions. However, combining raw data on species
calcification rate proved to be more challenging because even across the same species the reporting
of the calcification rates was highly variable. The use of different measuring approaches of
calcification rates while conducting OA studies generated data with divergent units that do not
allow for the intercomparison of data and results. As different studies for a single species could
not be combined, we chose to increase the number of studies and thus, the number of examined
species. Based on the response categories from the OA studies (Ries et al., 2009), our hypothesis
was that OAE will elucidate the same categories of responses, i.e. positive, negative and neutral.
Within each of the groups examined, multiple categories of predicted calcification response were
found. In this way, we demonstrated that it was possible to develop a useful framework for
assessing and predicting species-specific OAE responses that can delineate different responders,
identify species with greater OAE sensitivity and determine the thresholds where such negative
responses could happen.

**4.2 Synthesizing biological response under OAE identifies positive and negative responders**

The responses were summarized across three emerging groups of responses: positive, negative,
and neutral (Fig. 6). We observe species-specific variability at the species level, which is related
to various calcification mechanisms across the observed groups. The greatest variability upon
NaOH addition within each group in calcification rate was evident in corals, dinoflagellates,
foraminifera, gastropods and pteropods, where four to five different categories of responses were
found.
Positive responders (34%) show an increased calcification rate upon alkalinity addition, observed
within all functional groups besides annelids, coccolithophores and dinoflagellates. Corals mostly
have positive and neutral responses, suggesting that coral species would not be negatively
impacted during OAE field trials. This mostly positive response is validated by increased coral
calcification, shown for two coral species of *Acropora* and *Siderastre* in experiments conducted
by Palmer et al. (2022).
The metrics to evaluate the sensitivity of calcification rate for the negative responders (negative
linear and threshold) to alkalinity addition was based on the amount of alkalinity addition required
to halve the current calcification rate (Fig. 3; Tables 1, 2). The most negative responses were found
in dinoflagellates (6% of all species), algae and foraminifera (both 5% of all species). However,
these numbers are affected by the difference in data coverage per functional group. When
comparing the ratio of negative to positive and neutral responses, crustaceans and dinoflagellates
are expected to be most negatively affected. As such, these groups are one of the priorities for the
future OAE experimental work to determine at which TA:DIC negative response happens.
Dinoflagellates demonstrate negative response in 5 cases, 5 neutral responses and 0 positive (see
Table 1; Supplemental Fig. 4). The reason for negative response to OAE in this group is related to
the fact that their growth gets limited at higher pH, with further carbon limitation playing a role at
very high pH levels and low DIC concentration (Hansen et al., 2002; 2007). On the other hand,
crustaceans only demonstrated positive response in one study (Pansch et al., 2014), while
remaining results predict either negative or neutral response. While crustaceans are effective in
retaining homeostasis at lower pH, they might be less so at higher pH, which was shown in the
OA experiments by Ries et al. (2009) for three crustacean species (*Callinectes sapidus, Homarus*
*americanus, Penaeus plebejus*), confirmed in the OAE study by Cripps et al. (2013) in *Carcinus*
*maenas*. While studies are still lacking, physiological acid-base regulation at higher pH is
associated with higher costs (Cripps et al., 2013). Crustaceans show a disrupted acid–base balance,
evident through the increase in hemolymph pH, $K^+$, $Na^+$ ions and osmolality, coupled with a
decrease in extracellular $pCO_2$ and $HCO_3^-$, indicative of respiratory alkalosis (Truchot,
1984;1986). This is often associated with hyperventilation, the aim of which is to flush out the
hemolymph $CO_2$ to increase the affinity of oxygen uptake. However, while this might be a
temporary physiological relief it also implies energetic costs, potentially also for calcification.
For the neutral responders or groups with no significant correlation between calcification rates and
TA:DIC, it is somewhat uncertain to predict if such responses will be retained under OAE. While
parabolic responders show a physiologically understandable parabolic type of dose-response,
positioning the TA:DIC values where the thresholds occur is also highly species-specific and
potentially uncertain, meaning that it might depend on other environmental factors.
With respect to the coccolithophores, we note that this was the only group where data compilation
on calcification rate across the group was possible because the OA studies were conducted in a
more uniform way, using similar approaches, and reporting the result in the same units. When data
for *E. huxleyi* across the comparable studies was compiled (Barcelos-Ramos et al., 2010; Fiorini
et al., 2011; Iglesias-Rodrigues et al., 2008; Sciandra et al., 2003; Stoll et al., 2012; Richier et al.,
2011), a significant parabolic response was obtained (Table 1), although the goodness of fit was
fairly low ($R^2$=0.16). Despite lower $R^2$, we decided to use the compiled dataset because of the
increased statistical power. The parabolic response obtained aligns with Langer et al. (2006) and
also with the parabolic type responses found in the synthesis studies by Paul and Bach (2020) and
Bach et al. (2015). The threshold indicates the mechanisms of coccolithophore growth that are
driven by $CO_2$, which is shown to decline with alkalinity addition. The threshold based on all
studies for *E. huxleyi* combined was positioned at a TA:DIC of 1.46 ($\Omega_{ar}$ = 13.65, see Table 2),
which would be triggered at 850 µmol/kg of added NaOH and at a $pCO_2$ of 60 µatm.
Comparatively with the phytoplanktonic diatoms, such growth limitation is predicted at a $pCO_2$
amount at 100 µatm (Riebesell et al., 1993). It is important to note that when these studies were
analyzed individually, a mixture of different responses was observed. We emphasize the variability
within the coccolithophore responses, which are species-specific and inherently related to the
strain adaptation to their innate regional settings and dependent on a variety of other factors (Bach
et al., 2015; Gafar and Schultz, 2018), including the longevity of the species, the experimental
settings used in the study (e.g. nutrient-replete vs nutrient deficient conditions) and the presence
or absence of (un)suitable light conditions. Interestingly, for all the coccolithophore species other
than *E. huxleyi*, responses were neutral. For validation purposes, the results of our study could not
be compared, either because the calcification rates were not studied or the calcification units were
not comparable (e.g. Diner et al., 2015).
**4.3 Parameters impacting derivation of thresholds and their application**
We developed a set of species-specific thresholds in this study, with demonstrated application
across the global $\Omega_{ar}$ vs TA:DIC conditions (Table 2; Fig. 8). The range of alkalinity additions to
result in a threshold of 50% decline in calcification rate varied significantly between the species
and the type of response. The TA:DIC thresholds upon TA application ranged between 50 to 1400
µmol/kg of NaOH addition and 2250 to 6500 µmol/kg of $Na_2CO_3$ addition, and the $pH_T$ 9
thresholds averaged at 1200 µmol/kg of NaOH and 4700 µmol/kg of $Na_2CO_3$ for all species.
However, there are many parameters that impact threshold derivation and application, which we
discuss in greater detail.
First, we note that differences in experimental conditions for different species make it difficult to
directly compare different species' thresholds among each other. Instead, they are intended to
delineate sensitivity to alkalinity addition of individual species at given experimental conditions.
In the case that the lab experimental conditions mimic species' natural habitat, this threshold-
related sensitivity can be extrapolated to their natural habitats.
Second, we emphasize that the threshold application should not only consider the magnitude of
NaOH added, but also the duration or exposure time of the experimental study. As such, when
applying the thresholds to respective model outputs or observation data, both duration and
exposure time should be considered. For all the derived thresholds, we have added duration
exposure information to Table 2. Additional parameters that need to be included when applying
these thresholds are related to local temperature and salinity. The extracted threshold values are
calculated with the temperature and salinity from the experimental conditions, which means that
this threshold should not be applied to very different conditions without adjusting for salinity and
temperature.
Third, we assumed global surface ocean conditions to be standardized at a $pCO_2$ of 425 ppm and
a $pH_T$ of 8.1 as a control point for OAE compound additions. However, we note that in different
habitats, $pH_T$ 8.1 may not represent the baseline from where OAE should be considered adding,
because the average pH might be different. This means that the amount of TA required to reach a
certain threshold could vary and is dependent on the baseline carbonate chemistry at the site of
deployment and its variability. This is especially relevant in habitats with a lower baseline pH,
where more TA would need to be added for the threshold to be reached, meaning less negative
biological implications.

In addition, physical parameters of importance are related to the dilution effect, mixing, retention
capacity, as well as the rate of the equilibration effects of the air-sea $CO_2$ uptake (Ferderer et al.,
2022; He and Tyka, 2023; Schulz et al., 2023; Wang et al., 2023), because they determine relevant
exposure duration and the variability of carbonate chemistry parameters across spatial and vertical
depths. Therefore, to obtain the most accurate and regionally applicable threshold for the species
of interest, it is recommended that the baseline for OAE additions be determined based on local
conditions.
Lastly, if similar conditions as induced by the OAE field trial are present in the habitats that species
inhabit, it is more likely that the species might be pre-adapted to such conditions. However, if
species have not been exposed to such conditions, OAE might induce rapid change in conditions
and species exposure, which could be more challenging for the species. As such, it is worth
considering that OAE deployments could be, at least for the most sensitive species, carried out not
as a single high dosage deployment, but rather as a more continuous, lower dosage application.
This would eliminate the swings and maxima in conditions, while also allowing more time for
species acclimation or migration during the initial injection of the OAE deployment. Ultimately,
it is the combination of all these factors that creates baseline exposure conditions that are relevant
in the context of biological outcomes (Wang et al., 2023).
**4.4 Direction of laboratory OAE experiments should change to incorporate field conditions**
The lab OAE experiments that are being conducted right now are done under different conditions
than in the field. The former are conducted with the aim of gaining a wide-ranging empirical
response, which implies high treatment levels of OAE additions. However, biogeochemical model
outputs show that OAE-related concentrations at the injection site are high for a short-time, while
the realistic field dosing upon rapid dilution due to mixing is low. Wang et al. (2023) reported that
the nearfield maxima in the respective investigation area of the Bering Sea is to increase TA by
about 10 µmol/kg in the nearfield and by about 1 µmol/kg of NaOH in the farfield region. As such,
we should be more concerned about the threshold of exceedance occurring at the low NaOH
dosing, rather than at high NaOH additions, because these are more realistic and point to the most
sensitive species. As a result, we explicitly emphasize the importance of including much lower
additions of TA in the experimental treatment levels to better support biological understanding and
OAE application in the field. In addition, prior to the lab experiments it would be important to
identify what type of response is predicted in the experimental species. This is especially pertinent
for the groups for which OA experimental data is limited and skewed towards the lowest TA:DIC
ratio (Fig. 8; Supplemental Fig. 4).
What is needed urgently for the safe biological field trial experiments is a set of protocols that are
species-, habitat- and local conditions- specific, which would allow for comprehensive and
comparative risk analyses and threshold determination. As part of this, we also need to develop
regionally specific indicators for biological monitoring. Ideally, such biological and environmental
risk monitoring and assessment would be accompanied by the application of the physical mixing
models with site-specific biogeochemical processes (Ho et al., 2023; Fennel et al., 2023) that can
predict the maximum expected TA increase in the nearfield and farfield regions of the study site,
representing a more realistic exposure and better informing further experimental work.
**4.5 Validating OAE responses based on the mechanistically-derived calcification**
This study establishes the predictions of responses that relied upon empirical experimental studies.
A good alternative to validating the predicted responses is to use species-specific mechanistic
responses, a more accurate representation of responses compared to empirical studies. Here, we
conducted a subset synthesis study for the two species of coccolithophores, using the results from
this study and compared it to the literature-derived mechanistic responses where the responses are
described with a different set of carbonate chemistry parameters. We wanted to determine to what
extent mechanistic relationships can contribute to improved, i.e. more accurate and certain, OAE
predictions.
For *Emiliania huxleyi*, we used experimental TA and DIC data to calculate the $[HCO_3^-]$, $[H^+]$ and
$[CO_2]$ concentrations to be able to use the mechanistic rate equation from Bach et al. (2015). We
calculated and plotted the rate derived via mechanistic approach and applied linear, polynomial
(second-order) and exponential regressions and chose the best fit based on the lowest p-value,
using the same method as for our experimental calcification rate data regressions. Like the
mechanistic rate regression based on three carbonate chemistry parameters was a parabolic fit
(Bach et al., 2015), we also obtained the same fit using the experimental calcification rate data (see
Fig. 9). However, when using the same approach for another coccolithophore species *Calcidiscus*
*leptoporus* (Bach et al., 2015), our best fit did not align with the proposed mechanistic response;
instead, a non-significant relationship was obtained using experimental data (Supplemental Fig.
5). Such comparisons reveal species-specific relationships are likely dependent on a lot of
parameters, with one equation alone not being operable among different species from different
experiments or over varied regional settings.

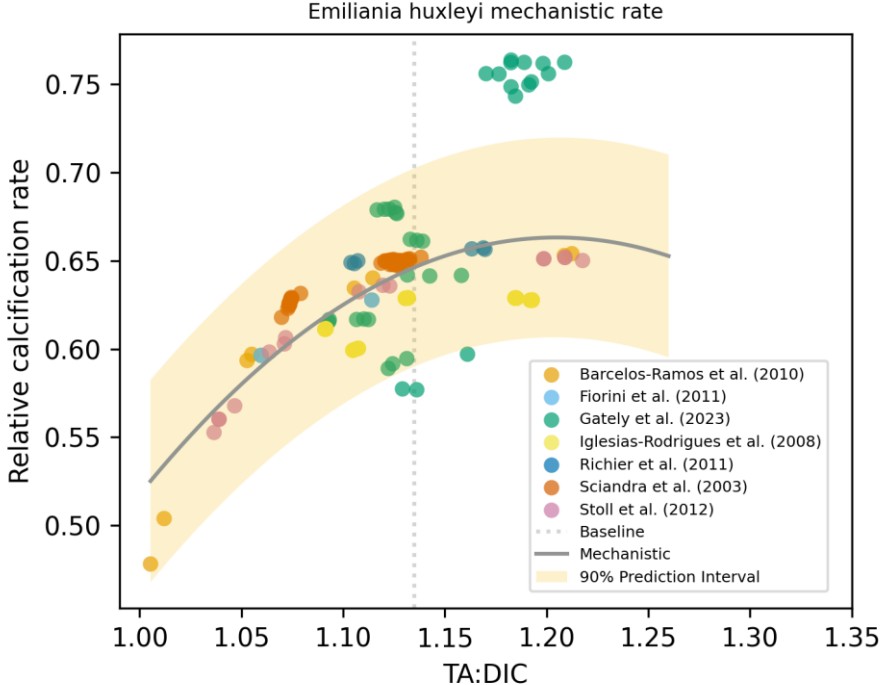


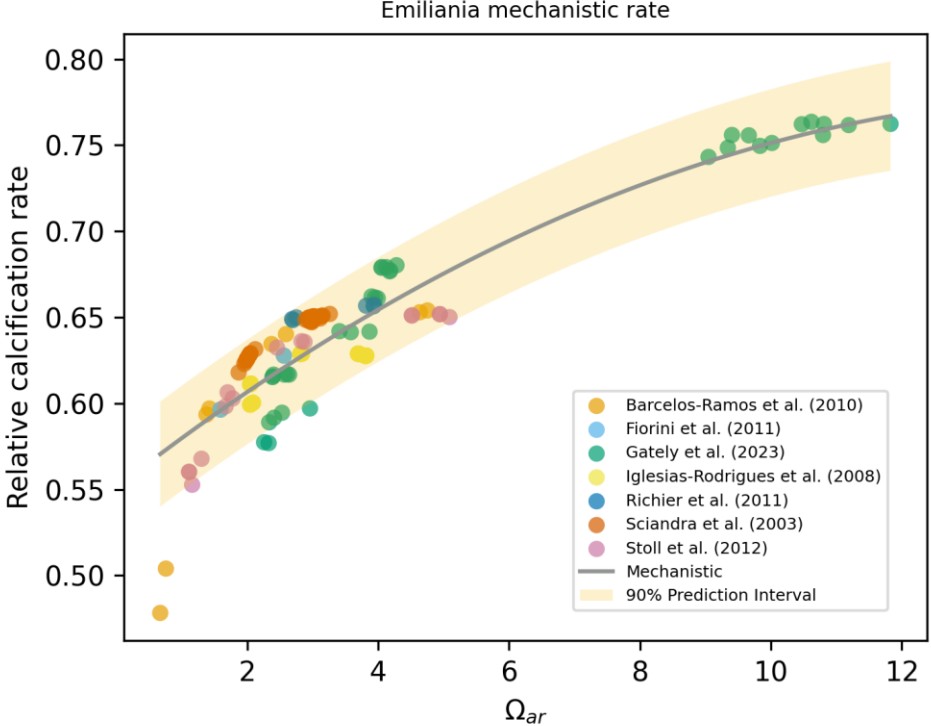


**Figure 9:** *Mechanistic rate equation and parameters (a = 9.56e-1, b = 7.04e-4 mol/kg, c = 2.1e6*

*kg/mol, d = 8.27e6 kg/mol) taken from Bach et al. (2015) and fitted using experimental data for*

*E. huxleyi (used data from the studies indicated in legend). Shading represents the 90%*

*prediction interval.*


For most of the species, such mechanistic relationships are not available. The substrate-to-
inhibitor ratio (SIR) (i.e. the bicarbonate ion to hydrogen ion concentration ratio) has often been
used to describe a calcification relationship based on a single parameter relationship. To
see if species rate group responses based on experimental data using TA:DIC vs calcification
rate could reproduce SIR relationships (TA:DIC vs SIR), we computed and plotted the
SIR ratio. This included calculating the bicarbonate and hydrogen ion concentrations in CO2SYS
using the experimental TA and DIC, for the mollusk, coral and coccolithophore groups, and
applying a best-fit regression model. We categorized these responses (using the categories
shown in Figure 2) and compared these SIR regressions to the respective best-fit regressions based
on the calcification rate responses from the experiments (shown in Table 1).

We found large differences between our calcification rate responses (based on TA:DIC vs
calcification) and the SIR-proposed mechanisms (Supplemental Fig. 6). For most of the
coccolithophore groups, the experimental rate regressions cannot be explained using SIR
mechanisms (i.e. the responses are different). Only in the case of *Calcidiscus leptoporus*,  the
experimental and mechanistic responses are the same (neutral). For mollusks, a third of the
mechanistic rate regressions based on the SIR agreed with the experimental calcification rate
regressions. The other two-thirds did not agree, especially for the studies with experimental
conditions of $\Omega_{ar} > 1$. For corals, the majority of the coral species (N=14) were classified as having
a linear positive mechanistic relationship when using SIR relationships. When comparing this to
our experimental rate regressions, we only found agreements between the experimental and
mechanistic regressions in 6 out of 18 species. It seems that SIR is a less common principle of
calcification and cannot be applied across a variety of species. It is likely that SIR might
insufficiently explain the multitude of biological processes involved in the calcification (e.g. how
carbon is provisioned or the ability to regulate calcifying fluid pH). Based on these results, the
general consensus is that the SIR ratio might actually tend to oversimplify species' calcification
rate responses. Ninokawa et al. (2024) and Li et al. (2023) emphasized that using only one
parameter to describe the calcification process is insufficient and strongly recommended using at
least two parameters for more accurate calcification predictions. Our findings agree with
Ninokawa et al. (2024), for example, we observe that using SIR relationships to successfully
describe calcification was limited to only a few species and that there are no generalizable patterns
that could be applicable across multiple groups.

Mechanistic models can offer better insights into calcification responses for some species,
especially when multiple environmental factors are accounted for, but they are not generally
applicable across taxa. Species-specific responses are influenced by unique biological and
physiological factors, which can lead to significant deviations between mechanistic and empirical
predictions. Therefore, mechanistic approaches will only provide valuable frameworks for species
with well-understood calcification processes. However, for many species covered in this study the
calcification process is not well-understood. By comparing mechanistic studies with experimental
data, we hoped to validate the predictive results of our experimental studies. This clearly delineates
a major gap in the mechanistic understanding of calcification so far, the lack of which significantly
limits our ability of ecological and biogeochemical predictions to OAE. As such, more research is
urgently needed on broader mechanistic understanding of calcification across different species,
and additionally, one parameter calcification processes should be replaced with more accurate and
comprehensive methods using two or three parameters.
**4.6 Unknowns about ecological and biogeochemical implications call for the precautionary**
**approach**
The value of calcification as the response proxy is indicative of organismal fitness, which directly
relates to OAE effects as harmful or beneficial for the species. From an ecological perspective, a
total of 26.0% negative responders demonstrates a potential for negative implications. In addition,
we note that this study did not include diatoms in the analyses, which are predicted to be negatively
impacted by carbonate-based OAE (Ferderer et al., 2022), leading to possible community-based
ecological shifts (Bach et al., 2019). The possibility of the ecological shifts should not be neglected
given the variety of the positive responders, understudied effects of OAE in non-calcifiers and
their relationship with the calcifiers through the grazing impact, and lastly, unknown and highly
unpredictable indirect effects. In addition, the inferences on the neutral responders should also
remain cautious.
From a biogeochemical perspective, it is reasonable to infer that OAE will introduce changes in
calcification rate across species, potentially resulting in changing the carbon export or carbonate
counter pump. Species-specific responses in major carbonate producers, i.e. coccolithophores,
foraminifera and pteropods show both negative and positive response, which could have strong
effects on biogeochemical fluxes (Riebesell et al., 2017; Bach et al., 2019). Increased calcification
could result in thicker and denser shells, contributing to faster sinking and increased carbonate
fluxes, while decreased calcification has the opposite effect. This could potentially induce changes
on the subsurface total alkalinity at intermediate and deeper depths in the water column, and
dissolution at or near the seafloor (Gehlen et al., 2011) or result in a potential feedback of increased
$CO_2$ flux to the atmosphere (Gattuso et al., 2021). The full scope of ecological and biogeochemical
shifts remains a high priority topic for future investigations and until these huge uncertainties are
resolved, we should exercise a precautionary principle in considering the next steps of OAE field
implementations.

**4.7 Potential confounding effects**

This study only considered the changes in carbonate chemistry due to the addition of NaOH and Na$_2$CO$_3$. However, other OAE feedstocks contain compounds that could induce biological toxicity due to the presence of trace metals (Ni, Cu, Ca, Si; Bach et al., 2019), as well as potential negative environmental impacts due to secondary precipitation (Hartmann et al., 2022; Moras et al., 2022). This study also did not focus on the sensitivity across different life stages, even though stage-specific sensitivities to OAE are expected based on previous OA results. Furthermore, we did include data from experimental lab and field studies that involve multiple stressors in their experimental designs. As such, an additional impact of warming, dissolved oxygen, and light intensity on the OAE-induced responses was not determined, although they could elicit different biological pathways than OAE alone or have additional confounding effects.

The synthesis of the experimental studies always includes implicit biases that are based on the published experimental studies, the range and species used, regional coverage and heterogeneity. Important consideration is the adaptation of the species used in the experimental studies because their calcification optimum might be pre-determined based on their local habitat conditions. Given that the baseline for the OAE-compound addition was chosen at the global current surface pH value, some of the thresholds might actually be lower than expected.

**4.8 Applications within the existing governmental regulations and the guiding principle**

Our results, especially related to the use of biological thresholds or NaOH dosing, could have wider applications, most notably with policy-management governmental regulations. For example, we calculated the amount of alkalinity addition required to reach the pH$_T$ threshold of 9, the maximum pH allowed by the US Environmental Protection Agency's for waste water entering the coastal ocean (see NPDES manual, 2010). To reach this threshold, 1200 µmol/kg of NaOH and 4700 µmol/kg of Na$_2$CO$_3$ was required on average for all species, with the lowest threshold reached at 750 µmol/kg NaOH and 2250 µmol/kg of Na$_2$CO$_3$ addition for *Amphibalanus improvisus*. This is a very high concentration, and the thresholds for most of the negative responders with identified thresholds (Table 2) will be exceeded far below the regulatory standards of pH$_T$ 9 (Table 2), especially if the exposure occurred over a duration period that matters for calcification and for the organism's physiological status. This case demonstrates discrepancy of the current chemical pH

regulation and associated biological effects, where safe biological limits are not considered, and
biological harm might not be prevented. Despite the fact that achieving such a high pH through
$NaOH/Na_2CO_3$ implementation is unlikely to occur in the field, such regulations currently do not
assure safety space for marine biota and they need to be urgently addressed.
**5. Conclusions and next steps**
Sufficient certainty in predicting biological responses reduces the risks and supports safe operating
space for OAE implementation and scaling up. Overall, given that almost 60% of examined species
showed non-neutral response (either positive or negative), this calls for careful implementation of
OAE until the safe operational temporal and spatial scales are identified and OA mitigation
measures are established. The goal of this study is to serve as a baseline for prioritizing
experimental and field OAE research and assess environmental risks. Such prioritization identifies
those species for which experimental work needs to be conducted first. This would involve species
with the greatest OAE-related sensitivity (negative responders), species with the greatest
uncertainty in response, as well as the species with very strong predicted positive response that
could potentially introduce a shift on the community level. In addition, it would also recognize the
species for which the existing knowledge is sufficient and there is less immediate need for the
OAE experiments. We hope that all presented tools provide guidance for the practicing and
regulatory community considering OAE field application within the safe limits.
It is important to emphasize that this study is the first comprehensive synthesis of the effects of
OAE. Ongoing updates and additional data would enhance its value, particularly when
complemented by further experimental research. Similar datasets on OA exist for various
biological parameters, including genetics, physiology, and survival data, as well as for non-
calcifying organisms. This availability allows for the exploration of ecological implications and
contributes to developing an ecosystem-based predictive risk assessment for OAE.
**Data availability**
No new data were generated during this study; all data were collected from previously published
studies. The compiled data is currently available on request. The Python code used for computing
baselines per species, conceptually adding alkalinity in the form of NaOH and Na2CO3, predicting
calcification rate response, visualizing data and computing thresholds is available in the GitHub
repository at https://github.com/hannavdmortel/OAE_calc_response (last access: 1 November
2024) and is archived on Zenodo at https://doi.org/10.5281/zenodo.14024442 (van de Mortel,
2024). PyCO2SYS v1.8.0 (Humphreys et al., 2022) was used to solve for the carbonate system,
with software available at https://doi.org/10.5281/zenodo.3744275 (Humphreys et al., 2023).
**Author contributions**
NB designed and conceptualized the research and wrote the first draft of the paper. HvdM collected
and curated data, conducted formal analyses and provided visualization. GP provided the analyses
using GLODAP data, and also provided visualizations and formal analyses. MGR has provided
formal statistical analyses and visuals. RAF and AD have provided insights, suggestions, and
generated discussion about specific parts of the paper. All have contributed to the writing of this
paper.

**Competing interests**
The contact author has declared that none of the authors has any competing interests.
**Financial support**
This study was funded by the NOAA NOPP project (mCRD 48914-2023 NOAA to AD, NB, and
RAF), with the title: mCDR 2023: Assessing chemical and biological implications of alkalinity
enhancement using carbonate salts obtained from captured $CO_2$ to mitigate negative effects of
ocean acidification and enable mCDR). This project also fully supported HvdM who worked on
the project as an external consultant. This work was supported by NOAA funding from the
Inflation Reduction Act and the Ocean Acidification Program (ROR ID: 100018228). NOAA's
Ocean Acidification Program supports this project on behalf of the National Oceanographic
Partnership Program (Award #NA23OAR0170516). HvdM has also been supported through the
Slovenian research Agency (ARRS J1-2468, N1-0359). This is PMEL contribution number 5621.

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
