# Peer review of "Assessment framework to predict sensitivity of marine calcifiers to ocean alkalinity"

_EGUsphere, 2024_

## Author Comment (AC1)

**RC2: ANONYMOUS REFEREE #2**

*Overall assessment:*

*This is a timely and significant contribution and I applaud the authors. This manuscript is ambitious, covering a range of relevant species for calcification rate responses to OAE, and will be of interest to biological oceanographers, ecologists and the wider carbon removal community and industry. The paper has a fantastic coverage of species and I really like the focus on functional groups; this is a sensible way to think about ecosystem response to OAE, in terms of what function each of these organisms play in the marine food web and carbon cycle more generally e.g. biological and inorganic carbon pumps. I really enjoyed reading the paper and thought it was generally well written and scoped. However, I have some major points raised below and also some more minor under "other comments". The conclusions of the study are solid - there should be more realistic manipulations of TA for responses of biological organisms in academic studies, and this is a fair interpretation of the data. Statistical analysis seems fair.*

We would sincerely like to thank the reviewer for recognizing our objectives and efforts, and for the useful feedback. Comments are addressed below.

*Major points for revision:*

1. *Better elucidation of link between TA:DIC and $\Omega_{ar}$ needed. For example, in the text introducing the rationale, it is just described as TA:DIC being "essentially" related to $\Omega_{ar}$, however this relationship is not explained in any depth and then in Figure 1, TA:DIC/$\Omega_{ar}$ is plotted on the x axis of the graphs, making an assumption that they are equivalent for those purposes. I think TA:DIC is a fair proxy for $\Omega_{ar}$, but should not be plotted in a way that misleads to suggest they are functionally equivalent and could be 1:1 swapped out for one another in graphs such as those shown in Figure 1. A lot of this links back to methodology for this relationship being used which is shown in Figure 3; perhaps moving this graph to earlier in the flow of the manuscript could help to clear up some of this confusion.*

**Response:** We have covered the explanation of the link between TA:DIC and $\Omega_{ar}$ in our response to reviewer #1, point 1. We have expanded on this in the manuscript as well in section 1.2, and therefore believe leaving figure 3 in the start of the Results section is reasonable. We have removed $\Omega_{ar}$ from the figure 1 x-axis.

2. *It seems questionable that GLODAP should have been used, the data logically to me should have come from the individual studies relevant and used in this study; I am unclear of the reasoning here and it does not seem well justified to be using global datasets for mapping some of these responses, but I note this justification is not well detailed in the text and could be a lack of explanation rather than a fundamental flaw in the design of the study.*

**Response:** This is explained in our response to reviewer 1, point 3.

3. *Not enough examination of what the actual impact of increased/decreased calcification rate would be in the manuscript. This is generally discussed in the context of biological responses to OAE, insinuating that any change is a bad change - this is not true - we have to think about this in the context of some organisms having decreased calcification under acidification scenarios, so some increase in calcification rate for these organisms could actually represent a bounce back to pre-acidification conditions. Overall, important to understand that any intentional manipulation of the earth system like OAE will have an impact, but the actual "negativity" of the impact needs to be a nuanced and detailed discussion, which I think this manuscript could benefit from significant additional discussion on this in the Discussion section of the manuscript. Places where this is particularly relevant: lines 562-564 and onwards.*

**Response:** We appreciate this valuable comment by the reviewer that helped to elucidate more nuanced discussion about the categories-specific calcification while taking into account the anthropogenic changes in carbonate chemistry since the pre-industrial times. We have done additional analyses of the pre-industrial calcification now and have amended the text in the manuscript to be more precise in addressing the complexity in responses. Below is the short *background* on how we have tackled this in our edits of the manuscript.

Anthropogenic CO2 uptake in the ocean resulted in lower pH and $\Omega_{ar}$, resulting in lowering of the calcification rates in the species, in particular the ones that are in our study were categorized as the positive responders (linear and threshold positive due to increase in TA-induced $\Omega_{ar}$). With respect to the changes since the pre-industrial times, the aim was to examine the difference in calcification between current and pre-industrial times, and to what extent NaOH would be able to compensate for this difference. We have attached two figures as examples of linear positive and threshold positive responders, in which we have clearly indicated the current and pre-industrial level of calcification to help visualize the text we provided (see Figure 5).

To understand this, we have done the following: first, we inferred the pre-industrial TA:DIC ratio of 1.16 (Feely et al., 2004) vs. a current TA:DIC of 1.12 and use the regression lines of TA:DIC vs calcification rate to calculate the corresponding calcification rates. Then we calculated calcification due to the addition of NaOH and $Na_2CO_3$ from the species-specific baselines (see Method section 2.4 for detailed explanation) for the positive responders. This is done using the principles of mass balance approach for the carbonate system via CO2SYS, where the carbonate system is calculated for each increment of NaOH or $Na_2CO_3$ added. The difference between the pre-industrial calcification and current, increased by the NaOH was assessed and compared. We observe the following:

- The average calcification rate in the species with the *threshold positive* rate was similar for the pre-industrial and current conditions, or with generally small difference between the pre-industrial and current conditions. This is likely because they retain max calcification rate even at the current rate and the changes since the pre-industrial did not induce calcification challenges yet.
- However, a very different case was evident for the linear positive calcifiers, where the current calcification rates are substantially lower compared to the pre-industrial calcification rates,

indicating that the calcification rates have been substantially compromised already in these species. An increase using NaOH would compensate for the calcification loss due to lowering of pH and aragonite saturation state since the pre-industrial times. Interestingly, we find a rather uniform pattern for most calcifiers where the difference between the pre-industrial and current conditions would be compensated if an addition of 50 to 150 µmol/kg NaOH was added. This amount is species specific, but at least this represents a wider range of NaOH that would be needed to revert back to the pre-industrial calcification. We also emphasize that for a much smaller amount of species (e.g. coral *Duncanopsammia axifuga* from Bove et al., 2020), we would need less than 50 µmol/kg NaOH to achieve pe-industrial calcification.

- This represents a substantial increase, translated into the pH and $\Omega_{ar}$ of an average increase of 0.09 for pH, and 0.54 for the aragonite. For these species, we could then indeed conceptualize that this NaOH addition would then allow these species to *first bounce back to the pre-acidification conditions*, and as such not induce competitive challenges in the community up to the values of NaOH added of 50-100 µmol/kg. Achieving such long-lasting increases of NaOH is currently not envisioned and as such, we are not expecting this as it represents huge carbonate chemistry amendments, meaning we do not envision competitive challenges to be an issue for the linear responders. We have amended our text in the discussion in view of these new findings.

[Figure]

[Figure]

[Figure]

**Figure 5:** *Conceptual diagram to show how experimental data (green dots), predicted values at various additions of alkalinity (stars), the regression line and prediction error margins are fitted for a given species a) linear positive (top) and exponential for threshold positive (bottom). The uncertainty interval indicates four standard deviations. The red line indicates zero net dissolution (calcification rate is equal to 0; dissolution rate = calcification rate). The blue dotted horizontal line indicates current calcification rate (TA:DIC = 1.12) and the gray dotted horizontal line indicates pre-industrial calcification rate (TA:DIC = 1.16).*

*Other comments:*

*Generally found the presentation of the different categories of response e.g. linear +, linear -, exponential +, exponential -, etc a bit confusing, particularly in Figure 6 and 7. I wonder if a box calling out exactly what fits into each of those categories and what they mean would be helpful for the reader to refer to.*

The following box will be added alongside the bar chart, and the color coding (green for positive, gray for neutral and red for negative) will be used in the bar chart to make it clearer.

[Figure]

**Figure 6:** *Box with color coding to clarify which responses are positive, neutral and negative.*

*line 483-485: not a sufficient explanation of the biological mechanisms involved in bicarbonate impact moderation in crustaceans*

Calcification in crustaceans is enabled through the process of regulating pH at the site of the calcification, by converting HCO3- to CO32-, allowing the crustaceans to operate over large pH ranges, with such strategy being especially successful at low pH. While this gives the crabs competitive advantage at ocean acidification conditions (lower pH, less carbonate ion available) compared to the other species that are using bicarbonate ions directly, this might not be advantageous under high pH conditions where such mechanisms are needed. Studies are lacking on regulating calcification at higher pH, but there is evidence of the metabolic costs associated with the physiological regulation at higher pH (Cripps et al., 2013). Crustaceans show a disrupted acid–base regulation upon the alkaline compound addition. Individuals exposed to elevated $Ca(OH)_2$ concentrations showed an increase in hemolymph pH, K+ Na+ and osmolality and a decrease in extracellular $PCO_2$, TCO2 and $HCO_3^-$. These are essential physiological extracellular acid-based parameters, which were all significantly affected. Such response is according to the authors (Cripps et al., 2013) indicative of respiratory alkalosis (see also Truchot, 1984, 1986), which is often associated with hyperventilation. Hyperventilation is associated with increased respiration with the aim to increase the flow of water over the gills, flushing out the hemolymph CO2, increasing the affinity of oxygen uptake of the hemocyanin. While this creates physiological favorable conditions for crabs, it also represents the physiological costs and potential metabolic composition of other processes, such as calcification. We fully acknowledge that this remains a hypothesis right now, however this might nevertheless explain the potential mechanism of a decline due to alkalinity addition. We will additionally expand on this in the manuscript.

*line 552: should read, such "a" guide*

Fixed.

*line 559: should be "better informing further experimental work"*

Fixed.

*line 585: when "the" grazing effects included - "the" should be removed*

Fixed.

*line 589, from "a" not the "the" biogeochemical perspective*

Fixed.

*line 600-601: NaOH is not a carbonate-based compound*

We have removed this, it now reads: 'This study only considered the changes in carbonate chemistry due to the addition of NaOH and $Na_2CO_3$.'

*line 605: "based" after OAE should be deleted*

Fixed.

*line 609: should be "were" not "was" used<<*

Fixed.

*line 618-619 - second part of clause not necessary, can be deleted*

Has been deleted.

*One of the major conclusions of the study is that the study is aiming to be used by the community to identify species that are at the largest risk of a "negative" response, or have the greatest uncertainties in their potential responses. However, nowhere in the paper is there a succinct summary of what those species are, in a box, or in a section, or a table. I think something like this would greatly improve the impact and applicability of the paper for the intended audience to use it, and the manuscript would be improved by this*

We believe Table 2 provides a clear overview of the negative responders, along with their thresholds. We aim to improve this table by translating the thresholds to pH and $\Omega_{ar}$ and provide the respective errors (RMSE).

*line 626: replace "the" with "further" experimental work*

Fixed.

*line 630: add "an" experimental framework*

Fixed.

---

## Author Comment (AC2)

**RC1: ANONYMOUS REFEREE #1**

*The study by Bednarsek et al utilizes available data from the ocean acidification literature to evaluate how marine calcifiers could respond to ocean alkalinity enhancement (OAE). The analysis takes a statistical approach. The key concern is their use of the TA:DIC metric, which is considered indicative for carbonate chemistry changes induced by OAE due to its correlation with Omega (concerns detailed below). The value and conceptual basis of TA:DIC is currently unclear or possibly not valid. Therefore the authors would need to use another metric or provide a much better justification for the use of TA:DIC that is found on more than a correlation with Omega. I also have several other comments that warrant attention.*

   1.
   - *The link between TA:DIC and OAE is perhaps not valid or at least not sufficiently well justified. In OA research, TA:DIC only changed because of increasing DIC. This, however, leads to other changes of relevant carbonate chemistry parameters (e.g. pH, CO2, HCO3-, CO32-) than a change in TA. There is no plausible explanation why the ratio of TA:DIC is a valid metric for OAE. The one argument made (control of TA:DIC on Omega) is not convincing so far because Omega itself may not be such a relevant metric for biotic calcium carbonate precipitation (although of course the relevant metric for abiotic precipitation/dissolution). If the authors cannot show that TA:DIC is indicative for OAE then the analysis of biological responses relative to this ratio is also not sound. Thus, demonstrating the conceptual validity of TA:DIC (much beyond a correlation with Omega) is crucial.*
   - *For the derivation of TA:DIC, it also needs to be considered with what state of OAE the concept correlates. Before equilibration with atmospheric CO2 or after?*
   - *One way forward could be correlation analyses of TA:DIC (where TA is left constant and DIC is varied, representative for OA), ideally using the data of the studies synthesised here. It could then be explored if TA:DIC is a useful metric for a specific transient state of OAE (e.g. unequilibrated or equilibrated with the atmosphere). However, if TA:DIC does not reflect carbonate chemistry changes of OAE more comprehensively (much beyond an Omega correlation) then the authors should use another metric to correlate their biological responses with.*
   - *Another question in this context is: If TA:DIC is used as a proxy for Omega, why wasn't Omega used in the first place?*
   - *Finally, ratios such as TA:DIC do not consider absolute concentrations, which is another potential weakness for a metric that has not been derived from physiological theory. This is particularly problematic because most of the data is sourced from ocean acidification research, which is looking at the other direction of the pH scale.*

**Response**: We appreciate the reviewer raising these points, which we aim to address in a combined way, because it will help us to target them much more comprehensively in the resubmitted manuscript. We indeed believe that we have not described these issues sufficiently in the previously submitted version of the manuscript and that we need to provide details and rationale for introducing the TA:DIC ratio and its

application to the experimental data. We provide a more structured background and reasoning on the importance of introducing the TA:DIC ratio and indicate where aragonite saturation state ($\Omega_{ar}$) falls short when conducting OAE experimental work. We explain how the TA:DIC ratio helps to simplify the carbonate system while conducting experimental work. We will add an extensive description of this in the resubmitted manuscript (we will create an additional Methods/Appendix section) where we can address this efficiently and comprehensively, particularly for those readers who need these guidelines when conducting experimental work.

For the background: whenever various OAE compounds are added to seawater so as to take up $CO_2$ from the atmosphere, the seawater carbonate chemistry changes in a multifaceted way affecting all acid-base species present in the seawater, resulting in all four of the commonly measured parameters (TA; DIC; $pH_T$; $pCO_2$) changing simultaneously. Understanding of such changes is absolutely essential for those biological experimentalists who are conducting biological assessments to assess potential OAE effects. However, assessment of the changes in the carbonate chemistry induced by OAE is neither intuitive nor straightforward; it requires detailed carbonate chemistry calculations. Nor is it clear as to what constitutes an appropriate "independent variable" that one can use to summarize the state of acid-base systems in a particular seawater when planning or reporting such biological responses. It is convenient when considering seawater acid-base chemistry to describe the system in terms of two of the four parameters noted above (as well as the salinity, temperature, and pressure), though strictly that is only true for a system where the both the total quantities of non-$CO_2$ acid-base systems, and the various equilibrium constants themselves can be inferred from S, T, & p alone, otherwise an additional piece of information is needed for each additional acid-base pair.

The approach we took was first to assume that such a restricted system was a reasonable approximation, and then to assume that TA and DIC were known for the various seawaters involved. (This pair has the added convenience that, when expressed as amount contents (moles per kilogram of seawater)), they are separately conservative to mixing, *i.e.*, the initial composition obtained by mixing a seawater (with known TA and DIC) with an OAE compound (whose chemical composition is itself known) can be estimated provided that the mixing ratio is known. We then chose to use the ratio of TA:DIC as a potential independent variable (an examination of the plots in Figure 1 below shows that the various isolines representing $pH_T$, $pCO_2$, and $\Omega_{ar}$ are approximately straight lines, *i.e.*, implying they correspond to a single value of the ratio TA:DIC).

Figure 1 shows the changes in the carbonate chemistry system inherent upon NaOH (black line) and $Na_2CO_3$ (dotted line) additions. Using such graphs of TA vs DIC (appropriate to a single salinity, temperature, and pressure) makes it easier to estimate compositional changes resulting from different alkaline additions in experimental settings. It is important to note that this estimate represents the initial state of the OAE conditions and is not representative of equilibrium conditions with respect to uptake of $CO_2$ from the atmosphere. That process would imply a further increase in DIC until the estimated $pCO_2$ equaled that of the atmosphere itself.

When we add NaOH, we increase TA only, and when we add $Na_2CO_3$ we increase TA and DIC at a 2:1 ratio (Figure 1a). With such additions, we then clearly understand how $\Omega_{ar}$ (Figure 1b) and $pCO_2$ change (Figure 1c), and how much of a change is required to bring the system back to equilibrium with respect to the atmosphere. We should also point out that if TA < 1000 µmol/kg and DIC < 500 µmol/kg, the isolines for $\Omega_{ar}$ are no longer straight, however, such conditions are rare in the ocean and thus not widely applicable.

[Figure]

**Figure 1:** *The effect of changes in TA and DIC on the properties of seawater (S= 34.7, T=20°C, [SiO$_2$] = 50 µmol/kg, [PO$_4^{3-}$] = 0.5 µmol/kg, TA = 2200 µmol/kg, DIC = 1950 µmol/kg), based on figures from* Schulz et al. (2023). *Pink dots (figures a-c) represent experimental TA and DIC data used in our meta-analysis, orange dots (figures d-f) represent GLODAP data for surface waters (0-50m depth). Subfigures show pH$_T$, aragonite saturation state and pCO$_2$. Calculations were carried out with the Python version of CO2SYS (*Humphreys et al., 2022*) using the stoichiometric dissociation constants for carbonic acid from* Sulpis et al. (2020), *for sulfuric acid by* Dickson et al. (1990) *and the total boron value from* Uppström (1974). *The solid black line indicates the effect of adding NaOH and the dashed black line indicates the effect of adding Na$_3$CO$_3$. This pair of lines can be translated on these plots so that its initial position moves elsewhere on these figures, to visualize different initial conditions for TA and for DIC (all other parameters used: S, T, p, Si$_T$, remain the same, as do the various equilibrium constants.*

In addition, TA:DIC is also an excellent proxy for the carbonate ion concentration in such plots. From a number of biological experimental studies, we know that carbonate ion concentration is the appropriate driver of the calcification process for many calcifying groups, although not all, rather than aragonite saturation state ($\Omega_{ar}$), which is proportional to carbonate ion concentration provided that the amount of calcium in the solution does not change. In that way, the TA:DIC ratio perhaps represents the calcification process better than $\Omega_{ar}$. Furthermore, using TA:DIC could also work for species in which other parameters drive the calcification, e.g. bicarbonate in autotrophic organisms, aragonite saturation state in bivalves

and H$^+$ flux in foraminifera. In that way, we encompass all the parameters that would otherwise influence the carbonate system and come up with a more straightforward way to express the experimental conditions, which would then enable easier comparisons among the experiments.

2. *The methods require a much more thorough description of what has been done. Some important steps are insufficiently clarified (see specific comments below).*

**Response:** We have extensively expanded the method section and the description of all important steps in the manuscript.

3. *What was the rationale for using GLODAP data to derive correlations between TA:DIC and Omega? Wouldn't it have been more reasonable to use data from the individual studies. Furthermore, how were temperature (salinity) differences between studies taken into account, which also affect Omega but not TA:DIC.*

**Response:** As suggested, we have additionally re-created Figure 8 from the manuscript where we plotted experimental data and derived thresholds. The graph shows comparable patterns of the TA:DIC vs $\Omega_{ar}$ correlation as also observed on the graph with GLODAP data. We show both images displayed in this response. We agree that this fits the context of deriving and using experimental data and explaining the implications. However, the graph with the experimental data shows a much greater spread of experimental data; this might be due to multiple causes (e.g. the use of wrong TA and DIC coupled values or the impact of temperature and salinity). We note that for each TA and DIC datapoint, the salinity and temperature specific to that data point were used to compute the saturation state of aragonite. This saturation state is plotted against TA:DIC in Figure 2. As such, displaying the TA:DIC relationship and threshold in the GLODAP context still remains valuable, also for the demonstration how the thresholds can be derived from the experimental data and be extrapolated in the field data across much larger TA:DIC gradients to demonstrate global distribution. The reasoning for extrapolating data and thresholds across the regional scales is based on the fact that we could apply the thresholds even for the regions for which we do not have sufficient or reliable data or experimental coverage, allowing to make the inferences about the OAE impact even in those regions. As such, we will continue using the graph with GLODAP data and thresholds, but with removed experimental data in the main part of the document, while we can add the graph with experimental data in the Supplementary material.

[Figure]

**Figure 2:** *On top is the GLODAP data, showing the correlation between TA:DIC and $\Omega_{ar}$. Below this is our experimental TA:DIC plotted against $\Omega_{ar}$, with the latter being computed using experimental TA and DIC, as well as the temperature and salinity specific to each data point. The GLODAP regression line is plotted on top to show that the trend is the same, but the experimental data has more scatter.*

4. *The statistical approach is enigmatic to me. I am unsure if it is insufficiently described or I am lacking statistical knowledge (quite possibly the latter). Hopefully the other reviewer has better understanding and provide a robust review.*

**Response:** Ries et al. (2009) use a similar method, where they chose 'the regression analysis that yielded the lowest square root of the mean squared error (RMSE) for a given species and that was statistically significant ($p \leq 0.05$)'. When applying this method to our data, parabolic and exponential regressions were always favored over linear regressions. When examining these regressions, we found that choosing the

best fit based on the lowest p-value yielded better fits, as this method prevents overfitting to noise in the data. Therefore, we chose final responses based on the lowest p-value, and not the RMSE. According to reviewer #2, the statistical analysis seems fair.

*Other comments:*

*Line 24: So far OAE has no relevance for climate change mitigation.*
Reword to: 'OAE is gaining prominence in its ability to mitigate climate change and ocean acidification.'
*Line 30: success or social license?*
Changed to: 'social license and success'.
*Line 40: No parabolic responses?*
Not sure what the reviewer means here, the parabolic response is mentioned.
*Line 43: What does realistic refer to here? That the conditions in most lab studies are not realistic wrt perturbation magnitude? Also, what would be a realistic perturbation. (I agree with your point but think this is not thoroughly backed up at this stage).*
This sentence has been removed to avoid confusion.
*Line 46: It is totally unclear at this stage what the TA:DIC ratio is a proxy for and why it is helpful. This must either be explained or taken out of the abstract.*
We have removed this from the abstract.
*Line 48: Unclear what framework you are referring to. TA:DIC? Needs specification.*
Added: 'framework based on TA, DIC and calcification rates that…'
*Line 69: CDRs is a weird plural. Removals?? Do you mean marine CDR methods?*
'CDRs' -> 'CDR'
*Line 78: Some OAE methods are well beyond concept stage (see Eisaman et al., 2023)*
Changed to 'despite mostly being in the concept stage', and Eisaman et al. (2023) added as a source.
*Line 100: adaptation or acclimation? (I think you mean the latter).*
'Adaptation' changed to 'acclimation'
*Line 109: Alkaline or "higher pH". Strictly speaking OA still investigated alkaline conditions wrt pH.*
'Alkaline' -> 'higher pH'
*Line 121: Unclear how a systematic framework should help here. Vague term.*
Changed to: 'This study aims at systematically evaluating species responses under OAE influence, placing them into a framework of categories based on calcification rate responses.'
*Line 139: The sentence implies that massive applications will happen anyway, in which case the environmental assessment before would implicitly have no influence on whether they are implemented.*
Changed to: 'delineate what experiments are most urgently needed to fill knowledge gaps before massive OAE field implementation can be considered.'
*Line 153: Based on what criterion were studies selected? Were all studies selected that were found by browsing? Or the first X hits?*
Added: 'For several functional groups data was easy to find (algae, corals, foraminifera, mollusks and dinoflagellates), so no new studies were added after 10 to 15 studies were found. Around five studies were found for the coccolithophore, crustacean, echinoderm and gastropod groups. Only one study was found for both pteropods and annelids.'

*Line 153: Were temperature differences between treatments in the OA studies considered?*

For CO2SYS calculations, a temperature of 20°C and the average salinity per species was used. To calculate the effect of this choice on our results, we computed CO2SYS calculations up until an addition of 500 μmol/kg for both NaOH and $Na_2CO_3$ for the maximum and minimum temperature per species, as well as the maximum and minimum salinity per species and compared this to the results for T = 20°C and average salinity. The largest difference for each combination of maximum/minimum salinity and maximum/minimum temperature compared to our original results for an NaOH addition of 500 μmol/kg. pH had an average uncertainty of 0.12 and a max of 0.35. For $\Omega_{ar}$, the average uncertainty was 0.30, with a max of 0.12.

*Line 242: Unclear how NaOH was added to the TA:DIC ratio.*

We believe this is thoroughly explained in lines 216 - 228.

*Line 242: Unclear why a parabolic response is by default a negative response. This requires additional justification.*

It is classified as a negative responder because upon a certain amount of alkalinity enhancement calcification rate will decrease. Rewritten as: 'Negative responders: species with predicted linear negative, parabolic and threshold negative response in calcification rate upon (a certain amount of) TA addition.'

*Figure 2 is unclear. Is this the entire dataset? Or a specific subset of data from various species. It is also unclear if each datapoint is a treatment level from an individual study.*

Legend updated to include green dots, labeled as 'Experimental data', and caption updated:

Figure 2: Conceptual diagram to show how experimental data (green dots), predicted values at various additions of alkalinity (stars), the regression line and prediction error margins are fitted for a given species a) linear positive; b) linear negative; c) parabolic; d) exponential for threshold positive; e) exponential for threshold negative. The uncertainty interval indicates four standard deviations. The red line indicates zero net dissolution (calcification rate is equal to 0; dissolution rate = calcification rate).

*Line 270: The "not strong" correlation between Omega and TA:DIC basically underscores that this metric is not representative of OAE.*

This line is saying the correlation is strong between $\Omega_{ar}$ and TA:DIC, and not strong for pH.

*Line 308: 98 or 96?*

This has been updated: due to new studies being added it is now 99 studies.

*Line 373: How do you convert a response observed as TA:DIC ratio into a delta_concentration above which thresholds are reached?*

We have the absolute concentrations for TA and DIC, not just the ratio. For each addition of NaOH or $Na_2CO_3$, we calculate the new TA, DIC, pH and $\Omega_{ar}$ using CO2SYS. TA and DIC changes are due to a simple mass balance, and are not affected by salinity or temperature. However, for pH and $\Omega_{ar}$ these are affected by temperature and salinity. To calculate threshold pH and $\Omega_{ar}$ we normalize for a temperature of 20°C, and use the average salinity per species. This is because most experiments were done at constant salinities, but at varying temperatures.

*Line 408: Several decades is exaggerated.*

'Several' has been removed.

*Line 412: while…while*

Changed to 'which will'.

*Line 446: An interesting question would be if the results here are consistent with predictions for species where predictions are possible due to mechanistic understanding.*

We thank the reviewer for this fundamental question that is based on the premise that we could predict calcification responses to perturbations in marine habitats if the mechanistic understanding of the unless calcification driver(s) were available. The premise is that the mechanistic relationships with identified carbonate chemistry driver(s) are available for species, which would make predicting calcification rate under various OAE scenarios feasible. Unfortunately, for most of the species, we still must rely on empirical, single-parameter relationships, for example saturation state, bicarbonate ion concentration, to substrate-to-inhibitor ratio (SIR) (i.e. the bicarbonate ion to hydrogen ion concentration ratio). In addition, more studies have shown that using a single parameter has significant limitations and also generates inaccurate changes in calcification to environmental changes, which was recently comprehensively elaborated in the study by Ninokawa et al. (2024), and supported by findings by Li et al. (2023). Both of the studies emphasize that using one parameter only insufficiently explains the calcification process, and as such, at least two parameters have to be taken into account for more accurate calcification predictions. Ultimately, it is important to note, that as a scientific community, we do not yet have an ultimate consensus on the carbonate chemistry drivers, and especially not on getting a generalizable pattern across the groups and more research efforts need to be dedicated to this.

To address the reviewer's comment, we used a few established mechanistic relationships of calcification for coccolithophores, bivalves and corals. We used these correlations and examined it against the experimental data compiled in our meta-analyses. We used experimental TA and DIC data to calculate the parameters represented in the mechanistic response.

For *Emiliania huxleyi*, we used the experimental TA and DIC data to calculate the [$HCO_3^-$], [ $H^+$] and [$CO_2$] concentrations. Using the mechanistic rate equation from Bach et al. (2015) and the sensitivity parameters for *Emiliania huxleyi* in table 1, we calculated and plotted the mechanistic rate. We applied linear, polynomial (second-order) and exponential regressions and chose the best fit based on the lowest p-value (same method as for our experimental calcification rate data regressions). Like the mechanistic rate regression, our experimental calcification rate also shows a parabolic relationship for *Emiliania huxleyi*.

[Figure]

***Figure 3:*** *Mechanistic rate equation and parameters (a = 9.56e-1, b = 7.04e-4 mol/kg, c = 2.1e6 kg/mol, d = 8.27e6 kg/mol) taken from* Bach et al. (2015) *and fitted using experimental data for E. huxleyi (used data from the studies indicated in legend).*

We also note that when using the proposed mechanistic relationship from Bach et al. (2015) for another coccolithophore species *Calcidiscus leptoporus*, the experimental values across the studies did not align with the proposed mechanistic relationship for this specific species; instead, a neutral relationship was obtained using experimental data. Given species difference in mechanistically explaining calcification rate between *Emiliania huxleyi* and *Calcidiscus leptoporus,* this reveals the fact that such relationships are likely very specific and dependent on a lot of parameters, with one equation not coved for different species from diverse regional settings.

Previously, some studies supported the use of the substrate-to-inhibitor ratio (SIR), i.e. the bicarbonate ion to hydrogen ion concentration ratio to mechanistically explain the calcification in marine calcifiers (Roleda et al., 2012; Cyronak et al., 2015; Fassbender et al. 2016). This relationship, which is tightly related to Ωar, attempted to better capture the calcification compared to Ω, however, in its recent form as a one-component parameter, it was evaluated as insufficient (Ninokawa et al., 2024). We have tested data from the experiments involving the mollusk, coral and coccolithophore groups against the SIR ratio.

[Figure]

***Figure 4:*** *Mechanistic rate equation and parameters (a = 2.31e5, b = 3.55e2 mol/kg, c = 2.19e5 kg/mol, d = 3.76e7 kg/mol) taken from* Bach et al. (2015) *and calculated using experimental data for 'Calcidiscus leptoporus' (used data from the studies indicated in legend).*

**For the coccolithophore group**, the experimental rate regressions cannot be explained using SIR mechanisms; most of the correlation predicts an insignificant response (p-value = 0.2;figure XY:). Reasons for these discrepancies could potentially be that SIR might insufficiently include various biological processes (e.g. how carbon is provisioned or the ability to regulate calcifying fluid pH), as well as salinity and temperature variations.

**For mollusk**s, half of the mechanistic rate regressions based on the SIR agreed with the experimental calcification rate regressions, the other half does not agree, especially for the studies with experimental conditions of $\Omega_{ar} > 1$ , as $\Omega_{ar} > 1$ was reported not to be an appropriate indicator of the calcifying environment (as suggested by Bach et al. (2015).

**For corals,** using SIR regressions, the majority of coral species (n = 21) were classified as having a linear positive mechanistic relationship. When comparing this to our experimental rate regressions, we only found agreements with the mechanistic regressions in 8 out of 21 species.

As such, we observe that using SIR relationships to successfully describe calcification was limited to only a few species and  there are no generalizable patterns that could be applicable across multiple groups, with our findings agreeing with Ninokawa et al., 2024.

*Line 466: The study by Albright did not show higher calcification rates but higher net calcification in a reef, which according to the authors could be due to reduced dissolution of the reef platform.*

The experiment by Albright has been removed as validation.

*Line 553: Unclear how the framework would be able to establish baseline conditions, which vary in time and space. If a new framework is proposed here then it should be spelled out much more thoroughly. It is currently very vague*

We have attempted to clarify this framework by elaborating further: 'should create a framework in which responses are predicted and categorized, establish initial baseline conditions, identify suitable risk analyses…'.

---

## Author Comment (AC3)

**CC1: SARAH COOLEY**

*This paper helps fill an emerging need: synthesizing what is known from other areas of study (here, ocean acidification) to shed light on the emerging topic of ocean alkalinity enhancement. The authors have done a prodigious amount of work to cull results from the literature that fit into their study parameters. Similar to Ries et al (2009), the paper seeks to identify different calcification responses by taxa when alkalinity is amended. But the synthesis leaves me with more questions than answers, in a way, because the most apt summary of the results is: "it's complicated" (see Fig. 5). Each taxonomic group includes anywhere from 1-5 different types of responses (linear -, linear +, etc.). I'm not particularly surprised by this, though, because since the Ries et al. proposed response curves kicked off this area of inquiry, numerous studies have pointed to metabolic and other complex physiological mechanisms being affected by ocean acidification, and calcification being kind of a metric describing changes in these other mechanisms. In 15 years of study, though, the community still has not really established whether more calcification leads to a biologically "better" outcome-- like greater survival or reproduction, or better quality as a food item for predators etc. So I feel as though calcification can't really be used as an indicator of biological harm/benefit from OAE. I don't agree that "winners and losers" can be identified given all these points.*

**Response:** We thank the reviewer for taking time to act as a public commenter on this. The line of thought is appreciated because it forces us to think more about how to make this work much more applicable, making sure we are addressing the gaps that would prevent the application of current work to the field or experimental settings. We believe that we have provided an added value to our manuscript by broadening the discussion and conducting some additional analyses related to the regulatory settings. We will integrate all the additional comments in the paper upon the next submission.

We only partially agree with the reviewer about calling the study 'complicated', we would rather characterized these responses as 'variable' while also showing the emerging patterns from this study. The steps of grouping the responses into the categories aim towards reducing the complexity and arranging the response into functional categories that are easier to handle. Using the response categorization with the aim of unifying, which was previously done in OA research, we now proposed a similar assessment that could unify different categories of species responses under OAE. Such work essentially leads to recognizing the most pertinent group of negatively impacted species that we are potentially most concerned about with the OAE field applications.

As per calcification response not being a 'good indicator of the OAE harm/benefit', we respectfully disagree with the reviewer. Calcification is a primary pathway of the organismal sensitivity to OA/OAE, which can act as an early warning response, and is directly implicated in growth and (abnormal) development across most of the marine calcifiers. Calcification also underlies the ecological success of numerous marine calcifiers. A large number of studies have strongly proven that OA affects the adaptability, growth or survival of larvae and juvenile economically important mollusks through the process of calcification and related acid-base balance (IPCC WGII, 2022; Vargas et al. 2022). Numerous studies also clearly show that the threshold for calcification occurs at similar pH/ $\Omega_{ar}$ values as the

thresholds for metabolic and energy metabolism processes (Lutier et al., 2022; Bednaršek et al., 2019; 2022). Furthermore, the implications of processes, both between and within biological levels are important. Ducker and Falkenberg (2020) recognized the importance the feedbacks moving between biological levels from "higher" to "lower" levels (e.g., compromised immune system affecting metabolic pathways) and within-level feedback cycles (e.g., reduced individual growth affecting energy expenditure affecting reduced individual growth. In the case of our study the changes in the process of calcification and development may subsequently affect metabolic or energetic rates (e.g. Stumpp et al., 2011; Ducker and Falkenberg, 2020). The level of calcification also directly addresses the level of susceptibility to predation, which impacts the mortality at the individual level but then leads to an altered size of the overall population. This summarizes the value of calcification as the proxy towards indicating organismal fitness, and this directly relates back to OAE effects as harmful or beneficial for the species.

We have refrained from labeling positive responders as being 'winners' and negative responders as being 'losers'. Such cautious wording is warranted given the high uncertainty of how the individual responses would play out in ecological interactions. However, we kept 'positive' and 'negative' responders, as this is clearly indicative of the individual responses to the carbonate chemistry change. This has led to the following changes in the title now reading 'Unifying framework for assessing sensitivity of marine calcifiers to ocean alkalinity enhancement categorizes responses and identifies biological thresholds - importance of precautionary principle' and section 4.2 header is: 'Synthesizing biological response under OAE additions identifies positive and negative responders. Discussion on whether positive responders can be considered winners is kept in section 4.5, paragraph 1.

*However, there are several points that this dataset and paper make that I wholeheartedly agree with: despite nearly 20 years of biological studies about OA, we don't have a clear idea of what will happen to marine organisms as a result of OAE; it is logical to anticipate there may be threshold responses due to the results of OA studies, and those thresholds may be lower than we had anticipated (also, by analogy, the OA community spent much effort on examining the role of natural and induced variability on OA, and it's reasonable to think variability may affect physiology at the other end of the pH scale also, but that's outside the scope of this study); there could be implications of OAE for the biological carbon pump that deserve more study; and making taxon-wide predictions or even place-based predictions about biological outcomes from OAE is nearly impossible. I would be hard pressed to use the outcomes of this study to identify what an ecological "safe operating space" for OAE experiments would be, as it pools species from many places into broad taxonomic groups and points to 40% of all species in the synthesis having neutral responses.*

*My recommendations for this paper include: a careful polishing for style and usage, because I saw a number of small errors and awkward phrasings that made the paper a bit harder to read; and revisions that "lean in" to the uncertainty and scatter that the synthesis uncovered. The authors are in a good position to show the magnitude of the challenge to draw comprehensive conclusions at this time from the OA literature regarding biological safety. I think by spending so much effort reporting details like the % of a taxon that had this vs. that response the results may be misinterpreted by people overly optimistic about whether OAE studies can be conducted in a biologically precautionary way. Data limitations and*

*experimental bounds from the OA literature both mean the existing data compiled probably aren't sufficient to provide community-wide guidance.*

**Response:** The review has not initially been conducted as a handbook for a safe operating space (for this, a lot more biological research needs to happen), but rather to propose the unifying assessment of species responses in three major categories. Such meta-analyses heavily rely on the use of data and knowledge generated during the OA research and allows for making more accurate predictions of biological responses under OAE. Having less uncertainty in the predictions related to OAE is of absolute importance, because of quick, multi-stakeholder (including industrial partners), advancements of the OAE field applications that are not followed quickly enough by the generated understanding of OAE effects from the scientific community.

This analysis helps us quickly recognize where potential concerns and gaps related to OAE implementations are. As such, hard numbers per functional group are less relevant, compared to the idea that OAE implementation would not necessarily mean a positive outcome for all the species. In fact, we strongly emphasized that in 60% of the cases we expect non-neutral responses that could imply some sort of ecological implications and we also caution that even neutral responses need to be tested in the lab to assure their neutrality. As such, we do not believe that our results are represented as overly optimistic, but rather imply a strong precautionary principle.

We were additionally challenged by the reviewer's conclusion of the irrelevance of the paper's results towards the community-wide guidance. As such, we have now added two sections in the body of the manuscript: first, a whole chapter on the efforts to be considered prior to conducting the field work. Second, we conducted further analyses, i.e. a case study that determines the guidance on predicting the suite of biological responses before the OAE field application. We have done this by taking regulatory standards and legislation on pH exceedance in account. We have considered the US Environmental Protection Agency's rule of not exceeding a pH of 9 for waste water entering the coastal ocean (see NPDES manual, 2010) and analyzed which species could be compromised because of added OAE because of the pH threshold exceedance in the relevant space-time exposure. Based on the analyzed calcification at pH, we infer that this pH 9 is not an issue for the positive responders, although it does create the conditions that favor the calcification during the exposure period to exceeding threshold. However, it could, e.g. if the exposure occurred over a duration period that matters for calcification, induce the challenges for the parabolic and negative responders, in particular for a few identified species that could have their calcification reduced, e.g. dinoflagellates (*Prorocentrum, Heterocapsa*) and foraminifera (*Marginopora*). We believe that both of these added components greatly increased the guidance for the community considering OAE field application.

---

## Author Response (AR2)

Corrections for the suggested changes from the Editor from 2nd October 2024

*We hereby acknowledge that we have addressed all the minor comments specified below. Our responses are in italics.*

*Sincerely,*

*Nina Bednarsek, on behalf of all the co-authors*

Minor comments:

L: 138: NaOH (black line), Na2CO3 (dotted), white in Fig. 1, please adjust- *Done*

L257: Sentence, please modify: "...whatever was the best(-fit) a determined?..." - D*one*

L276: Please check the sentence. - *Corrected*

L479: Please check the sentence "...understand the extent can OAE offset..." - *Corrected*

L837: Please check the sentence "We have added NaOh study until..." - *Corrected*

Fig. 2: please add information about the grey shade; prediction error?

*We have added this info.*

Fig. 3: Legend is very small, and it is hard to distinguish some of the symbols, e.g. 10 and 300 micromol addition of NaOH. Are 50 micromol steps really necessary here? There is no x and y-axis scale, as in the previous version. Why? Please include the scale again, as this is showing real data and not only a theoretical approach as in Fig. 2.

*We have corrected the figure.*

Figure 7: remove y= and x= . *Corrected*

Fig. 8: What are the thresholds lines (TA:DIC?), please specify, is it necessary to show all the different vertical lines or can this information be reduced?; remove y= and x= .

*We have now significantly reduce the number of thresholds and given that this now does not cause the redundancy and overcrowding of the figure, we have opted to keep it in as we*

*think it is informative and quick to grasp for the readership.  We hope you agree with our choice.*